# Graph Clustering With Missing Data : Convex Algorithms and Analysis

**Ramya Korlakai Vinayak, Samet Oymak, Babak Hassibi**
Department of Electrical Engineering
California Institute of Technology, Pasadena, CA 91125
{ramya, soymak}@caltech.edu, hassibi@systems.caltech.edu

## Abstract

We consider the problem of finding clusters in an unweighted graph, when the graph is partially observed. We analyze two programs, one which works for dense graphs and one which works for both sparse and dense graphs, but requires some a priori knowledge of the total cluster size, that are based on the convex optimization approach for low-rank matrix recovery using nuclear norm minimization. For the commonly used Stochastic Block Model, we obtain *explicit* bounds on the parameters of the problem (size and sparsity of clusters, the amount of observed data) and the regularization parameter characterize the success and failure of the programs. We corroborate our theoretical findings through extensive simulations. We also run our algorithm on a real data set obtained from crowdsourcing an image classification task on the Amazon Mechanical Turk, and observe significant performance improvement over traditional methods such as k-means.

## 1 Introduction

Clustering [1] broadly refers to the problem of identifying data points that are similar to each other. It has applications in various problems in machine learning, data mining [2, 3], social networks [4–6], bioinformatics [7, 8], etc. In this paper we focus on graph clustering [9] problems where the data is in the form of an unweighted graph. Clearly, to observe the entire graph on $n$ nodes requires $\binom{n}{2}$ measurements. In most practical scenarios this is infeasible and we can only expect to have *partial observations*. That is, for some node pairs we know whether there exists an edge between them or not, whereas for the rest of the node pairs we do not have this knowledge. This leads us to the problem of clustering graphs with *missing data*.

Given the adjacency matrix of an *unweighted graph*, a cluster is defined as a set of nodes that are densely connected to each other when compared to the rest of the nodes. We consider the problem of identifying such clusters when the input is a partially observed adjacency matrix. We use the popular *Stochastic Block Model* (SBM) [10] or *Planted Partition Model* [11] to analyze the performance of the proposed algorithms. SBM is a random graph model where the edge probability depends on whether the pair of nodes being considered belong to the same cluster or not. More specifically, the edge probability is higher when both nodes belong to the same cluster. Further, we assume that each entry of the adjacency matrix of the graph is observed independently with probability $r$. We will define the model in detail in Section 2.1.

### 1.1 Clustering by Low-Rank Matrix Recovery and Completion

The idea of using convex optimization for clustering has been proposed in [12–21]. While each of these works differ in certain ways, and we will comment on their relation to the current paper in Section 1.3, the common approach they use for clustering is inspired by recent work on low-rank matrix recovery and completion via regularized nuclear norm (trace norm) minimization [22–26].

In the case of unweighted graphs, an ideal clustered graph is a union of disjoint cliques. Given the adjacency matrix of an unweighted graph with clusters (denser connectivity inside the clusters compared to outside), we can interpret it as an ideal clustered graph with missing edges inside the clusters and erroneous edges in between clusters. Recovering the low-rank matrix corresponding to the disjoint cliques is equivalent to finding the clusters.

We will look at the following well known convex program which aims to recover and complete the low-rank matrix ($\mathbf{L}$) from the partially observed adjacency matrix ($\mathbf{A}^{obs}$):

**Simple Convex Program:**

$$\underset{\mathbf{L},\mathbf{S}}{\text{minimize}} \ \|\mathbf{L}\|_\star + \lambda\|\mathbf{S}\|_1 \tag{1.1}$$

subject to

$$1 \geq \mathbf{L}_{i,j} \geq 0 \ \text{ for all } i,j \in \{1, 2, \dots n\} \tag{1.2}$$

$$\mathbf{L}^{obs} + \mathbf{S}^{obs} = \mathbf{A}^{obs} \tag{1.3}$$

where $\lambda \geq 0$ is the regularization parameter, $\|.\|_\star$ is the nuclear norm (sum of the singular values of the matrix), and $\|.\|_1$ is the $l_1$-norm (sum of absolute values of the entries of the matrix). $\mathbf{S}$ is the sparse error matrix that accounts for the missing edges inside the clusters and erroneous edges outside the clusters on the observed entries. $\mathbf{L}^{obs}$ and $\mathbf{S}^{obs}$ denote entries of $\mathbf{L}$ and $\mathbf{S}$ that correspond to the observed part of the adjacency matrix.

Program 1.1 is very simple and intuitive. Further, it does not require any information other than the observed part of the adjacency matrix. In [13], the authors analyze Program 1.1 without the constraint (1.2). While dropping (1.2) makes the convex program less effective, it does allow [13] to make use of low-rank matrix completion results for its analysis. In [16] and [21], the authors analyze Program 1.1 when the entire adjacency matrix is observed. In [17], the authors study a slightly more general program, where the regularization parameter is different for the extra edges and the missing edges. However, the adjacency matrix is completely observed.

It is not difficult to see that, when the edge probability inside the cluster is $p < 1/2$, that (as $n \to \infty$) Program 1.1 will return $\mathbf{L}^0 = 0$ as the optimal solution (since if the cluster is not dense enough it is more costly to complete the missing edges). As a result our analysis of Program 1.1, and the main result of Theorem 1, assumes $p > 1/2$. Clearly, there are many instances of graphs we would like to cluster where $p < 1/2$. If the total size of the cluster region (i.e, the total number of edges in the cluster, denoted by $|\mathcal{R}|$) is known, then the following convex program can be used, and can be shown to work for $p < 1/2$ (see Theorem 2).

**Improved Convex Program:**

$$\underset{\mathbf{L},\mathbf{S}}{\text{minimize}} \ \|\mathbf{L}\|_\star + \lambda\|\mathbf{S}\|_1 \tag{1.4}$$

subject to

$$1 \geq \mathbf{L}_{i,j} \geq \mathbf{S}_{i,j} \geq 0 \ \text{ for all } i,j \in \{1, 2, \dots n\} \tag{1.5}$$

$$\mathbf{L}_{i,j} = \mathbf{S}_{i,j} \ \text{ whenever } \mathbf{A}^{obs}_{i,j} = 0 \tag{1.6}$$

$$\text{sum}(\mathbf{L}) \geq |\mathcal{R}| \tag{1.7}$$

As before, $\mathbf{L}$ is the low-rank matrix corresponding to the ideal cluster structure and $\lambda \geq 0$ is the regularization parameter. However, $\mathbf{S}$ is now the sparse error matrix that accounts only for the missing edges inside the clusters on the observed part of adjacency matrix. [16] and [19] study programs similar to Program 1.4 for the case of a completely observed adjacency matrix. In [19], the constraint 1.7 is a strict equality. In [15] the authors analyze a program close to Program 1.4 but without the $l_1$ penalty.

If $\mathcal{R}$ is not known, it is possible to solve Problem 1.4 for several values of $\mathcal{R}$ until the desired performance is obtained. Our empirical results reported in Section 3, suggest that the solution is not very sensitive to the choice of $\mathcal{R}$.

## 1.2 Our Contributions

- We analyze the Simple Convex Program 1.1 for the SBM with partial observations. We provide *explicit bounds* on the regularization parameter as a function of the parameters of the SBM, that

characterizes the success and failure conditions of Program 1.1 (see results in Section 2.2). We show that clusters that are either too small or too sparse constitute the bottleneck. Our analysis is helpful in understanding the **phase transition** from failure to success for the simple approach.

- We also analyze the Improved Convex Program 1.4. We explicitly characterize the conditions on the parameters of the SBM and the regularization parameter for successfully recovering clusters using this approach (see results in Section 2.3).
- Apart from providing theoretical guarantees and corroborating them with simulation results (Section 3), we also apply Programs 1.1 and 1.4 on a real data set (Section 3.3) obtained by crowd-sourcing an image labeling task on Amazon Mechanical Turk.

## 1.3 Related Work

In [13], the authors consider the problem of identifying clusters from partially observed unweighted graphs. For the SBM with partial observations, they analyze Program 1.1 without constraint (1.2), and show that under certain conditions, the minimum cluster size must be at least $\mathcal{O}(\sqrt{n(\log(n))^4/r})$ for successful recovery of the clusters. Unlike our analysis, the exact requirement on the cluster size is not known (since the constant of proportionality is not known). Also they do not provide conditions under which the approach fails to identify the clusters. Finding the explicit bounds on the constant of proportionality is critical to understanding the phase transition from failure to successfully identifying clusters.

In [14–19], analyze convex programs similar to the Programs 1.1 and 1.4 for the SBM and show that the minimum cluster size should be at least $\mathcal{O}(\sqrt{n})$ for successfully recovering the clusters. However, the exact requirement on the cluster size is not known. Also, they do not provide explicit conditions for failure, and except for [16] they do not address the case when the data is missing.

In contrast, we consider the problem of clustering with missing data. We explicitly characterize the constants by providing bounds on the model parameters that decide if Programs 1.1 and 1.4 can successfully identify clusters. Furthermore, for Program 1.1, we also explicitly characterize the conditions under which the program fails.

In [16], the authors extend their results to partial observations by scaling the edge probabilities by $r$ (observation probability), which will *not* work for $r < 1/2$ or $1/2 < p < 1/2r$ in Program 1.1 . [21] analyzes Program 1.1 for the SBM and provides conditions for success and failure of the program when the entire adjacency matrix is observed. The dependence on the number of observed entries emerges non-trivially in our analysis. Further, [21] does not address the drawback of Program 1.1, which is $p > 1/2$, whereas in our work we analyze Program 1.4 that overcomes this drawback.

# 2 Partially Observed Unweighted Graph

## 2.1 Model

**Definition 2.1** (Stochastic Block Model). *Let* $\mathbf{A} = \mathbf{A}^T$ *be the adjacency matrix of a graph on* $n$ *nodes with* $K$ *disjoint clusters of size* $n_i$ *each,* $i = 1, 2, \cdots, K$. *Let* $1 \geq p_i \geq 0$, $i = 1, \cdots, K$ *and* $1 \geq q \geq 0$. *For* $l > m$,

$$\mathbf{A}_{l,m} = \begin{cases} 1 \ w.p. \ \ p_i, & \text{if both nodes } l, m \text{ are in the same cluster } i. \\ 1 \ w.p. \ \ q, & \text{if nodes } l, m \text{ are not in the same cluster.} \end{cases} \quad (2.1)$$

If $p_i > q$ for each $i$, then we expect the density of edges to be higher inside the clusters compared to outside. We will say the random variable $Y$ has a $\Phi(r, \delta)$ distribution, for $0 \leq \delta, r \leq 1$, written as $Y \sim \Phi(r, \delta)$, if

$$Y = \begin{cases} 1, & \text{w.p.} \ \ r\delta \\ 0, & \text{w.p.} \ \ r(1-\delta) \\ *, & \text{w.p.} \ \ (1-r) \end{cases}$$

where $*$ denotes unknown.

**Definition 2.2** (Partial Observation Model). *Let* $\mathbf{A}$ *be the adjacency matrix of a random graph generated according to the Stochastic Block Model of Definition 2.1. Let* $0 < r \leq 1$. *Each entry of*

the adjacency matrix $\mathbf{A}$ is observed independently with probability $r$. Let $\mathbf{A}^{obs}$ denote the observed adjacency matrix. Then for $l > m$: $(\mathbf{A}^{obs})_{l,m} \sim \Phi(r, p_i)$ if both the nodes $l$ and $m$ belong to the same cluster $i$. Otherwise, $(\mathbf{A}^{obs})_{l,m} \sim \Phi(r, q)$.

## 2.2 Results : Simple Convex Program

Let $[n] = \{1, 2, \cdots, n\}$. Let $\mathcal{R}$ be the union of regions induced by the clusters and $\mathcal{R}^c = [n] \times [n] - \mathcal{R}$ its complement. Note that $|\mathcal{R}| = \sum_{i=1}^{K} n_i^2$ and $|\mathcal{R}^c| = n^2 - \sum_{i=1}^{K} n_i^2$. Let $n_{\min} := \min_{1 \le i \le K} n_i$, $p_{min} := \min_{1 \le i \le K} p_i$ and $n_{\max} := \max_{1 \le i \le K} n_i$.

The following definitions are important to describe our results.

- Define $\mathbf{D}_i := n_i \, r \, (2p_i - 1)$ as the **effective density** of cluster $i$ and $\mathbf{D}_{\min} = \min_{1 \le i \le K} \mathbf{D}_i$.

- $\gamma_{\text{succ}} := \max_{1 \le i \le K} 2r\sqrt{n_i}\sqrt{2(\frac{1}{r} - 1) + 4\left(q(1-q) + p_i(1-p_i)\right)}$ and $\gamma_{\text{fail}} := \sum_{i=1}^{K} \frac{n_i^2}{n}$

- $\mathbf{\Lambda}_{\text{succ}}^{-1} := 2r\sqrt{n}\sqrt{\frac{1}{r} - 1 + 4q(1-q)} + \gamma_{\text{succ}}$ and $\mathbf{\Lambda}_{\text{fail}}^{-1} := \sqrt{rq(n - \gamma_{\text{fail}})}$.

We note that the thresholds, $\mathbf{\Lambda}_{\text{succ}}$ and $\mathbf{\Lambda}_{\text{fail}}$ depend only the parameters of the model. Some simple algebra shows that $\mathbf{\Lambda}_{\text{succ}} < \mathbf{\Lambda}_{\text{fail}}$.

**Theorem 1** (Simple Program). *Consider a random graph generated according to the Partial Observation Model of Definition* (2.2) *with K disjoint clusters of sizes $\{n_i\}_{i=1}^K$, and probabilities $\{p_i\}_{i=1}^K$ and q, such that $p_{min} > \frac{1}{2} > q > 0$. Given $\epsilon > 0$, there exists positive constants $c_1', c_2'$ such that,*

1. *If $\lambda \ge (1 + \epsilon)\mathbf{\Lambda}_{fail}$, then Program* 1.1 *fails to correctly recover the clusters with probability $1 - c_1' \exp(-c_2'|\mathcal{R}^c|)$.*

2. *If $0 < \lambda \le (1 - \epsilon)\mathbf{\Lambda}_{succ}$,*

    - *If $\mathbf{D}_{\min} \ge (1 + \epsilon)\frac{1}{\lambda}$, then Program* 1.1 *succeeds in correctly recovering the clusters with probability $1 - c_1' n^2 \exp(-c_2' n_{\min})$.*
    - *If $\mathbf{D}_{\min} \le (1 - \epsilon)\frac{1}{\lambda}$, then Program* 1.1 *fails to correctly recover the clusters with probability $1 - c_1' \exp(-c_2' n_{\min})$.*

**Discussion:**

1. Theorem 1 characterizes the success and failure of Program 1.1 as a function of the regularization parameter $\lambda$. In particular, if $\lambda > \mathbf{\Lambda}_{\text{fail}}$, Program 1.1 fails with high probability. If $\lambda < \mathbf{\Lambda}_{\text{succ}}$, Program 1.1 succeeds with high probability *if and only if* $\mathbf{D}_{\min} > \frac{1}{\lambda}$. However, Theorem 1 has nothing to say about $\mathbf{\Lambda}_{\text{succ}} < \lambda < \mathbf{\Lambda}_{\text{fail}}$.

2. **Small Cluster Regime:** When $n_{\max} = o(n)$, we have $\mathbf{\Lambda}_{\text{succ}}^{-1} = 2r\sqrt{n}\sqrt{\left(\frac{1}{r} - 1 + 4q(1-q)\right)}$. For simplicity let $p_i = p$, $\forall\, i$, which yields $\mathbf{D}_{\min} = n_{\min} r(2p-1)$. Then $\mathbf{D}_{\min} > \mathbf{\Lambda}_{\text{succ}}^{-1}$ implies,

$$n_{\min} > \frac{2\sqrt{n}}{2p-1}\sqrt{\left(\frac{1}{r} - 1 + 4q(1-q)\right)}, \tag{2.2}$$

   giving a lower bound on the minimum cluster size that is sufficient for success.

## 2.3 Results: Improved Convex Program

The following definitions are critical to describe our results.

- Define $\tilde{\mathbf{D}}_i := n_i \, r \, (p_i - q)$ as the effective density of cluster $i$ and $\tilde{\mathbf{D}}_{\min} = \min_{1 \le i \le K} \tilde{\mathbf{D}}_i$.

- $\tilde{\gamma}_{\text{succ}} := 2 \max_{1 \le i \le K} r\sqrt{n_i}\sqrt{(1 - p_i)(\frac{1}{r} - 1 + p_i) + (1 - q)(\frac{1}{r} - 1 + q)}$

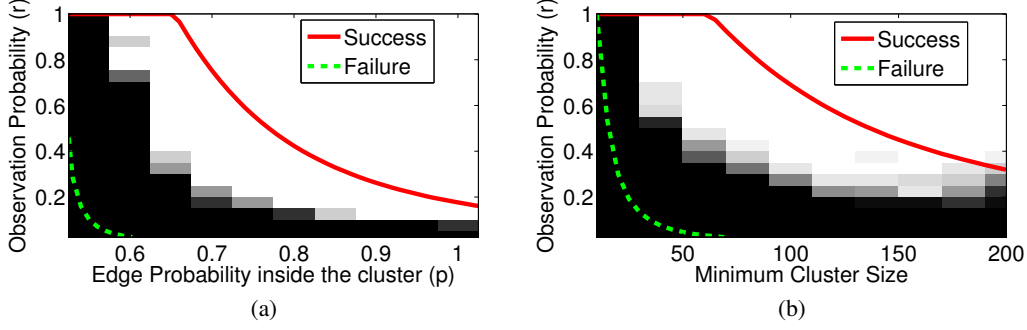

Figure 1: Region of success (white region) and failure (black region) of Program 1.1 with $\lambda = 1.01\mathbf{D}_{\min}^{-1}$. The solid red curve is the threshold for success ($\lambda < \Lambda_{\text{succ}}$) and the dashed green line which is the threshold for failure ($\lambda > \Lambda_{\text{fail}}$) as predicted by Theorem 1.

- $\tilde{\mathbf{\Lambda}}_{\text{succ}}^{-1} := 2r\sqrt{n}\sqrt{(\frac{1}{r} - 1 + q)(1 - q)} + \tilde{\gamma}_{\text{succ}}.$

We note that the threshold, $\tilde{\mathbf{\Lambda}}_{\text{succ}}$ depends only on the parameters of the model.

**Theorem 2** (Improved Program). *Consider a random graph generated according to the Partial Observation Model of Definition 2.2, with $K$ disjoint clusters of sizes $\{n_i\}_{i=1}^{K}$, and probabilities $\{p_i\}_{i=1}^{K}$ and $q$, such that $p_{min} > q > 0$. Given $\epsilon > 0$, there exists positive constants $c_1', c_2'$ such that: If $0 < \lambda \leq (1 - \epsilon)\tilde{\mathbf{\Lambda}}_{succ}$ and $\tilde{\mathbf{D}}_{\min} \geq (1 + \epsilon)\frac{1}{\lambda}$, then Program 1.4 succeeds in recovering the clusters with probability $1 - c_1' n^2 \exp(-c_2' n_{\min})$.*

**Discussion:**[1]

1. Theorem 2 gives a sufficient condition for the success of Program 1.4 as a function of $\lambda$. In particular, for any $\lambda > 0$, we succeed if $\tilde{\mathbf{D}}_{\min}^{-1} < \lambda < \tilde{\mathbf{\Lambda}}_{\text{succ}}$.

2. **Small Cluster Regime:** When $n_{\max} = o(n)$, we have $\tilde{\mathbf{\Lambda}}_{\text{succ}}^{-1} = 2r\sqrt{n}\sqrt{(\frac{1}{r} - 1 + q)(1 - q)}$. For simplicity let $p_i = p, \forall i$, which yields $\tilde{\mathbf{D}}_{\min} = n_{\min}r(p - q)$. Then $\tilde{\mathbf{D}}_{\min} > \tilde{\mathbf{\Lambda}}_{\text{succ}}^{-1}$ implies,

$$n_{\min} > \frac{2\sqrt{n}}{p - q}\sqrt{\left(\frac{1}{r} - 1 + q\right)(1 - q)}, \tag{2.3}$$

which gives a lower bound on the minimum cluster size that is sufficient for success.

3. $(p, q)$ **as a function of** $n$**:** We now briefly discuss the regime in which cluster sizes are large (i.e. $\mathcal{O}(n)$) and we are interested in the parameters $(p, q)$ as a function of $n$ that allows proposed approaches to be successful. Critical to Program 1.4 is the constraint (1.6): $\mathbf{L}_{i,j} = \mathbf{S}_{i,j}$ when $\mathbf{A}_{i,j}^{obs} = 0$ (which is the only constraint involving the adjacency $\mathbf{A}^{obs}$). With missing data, $\mathbf{A}_{i,j}^{obs} = 0$ with probability $r(1 - p)$ inside the clusters and $r(1 - q)$ outside the clusters. Defining $\hat{p} = rp + 1 - r$ and $\hat{q} = rq + 1 - r$, the number of constraints in (1.6) becomes statistically equivalent to those of a *fully observed* graph where $p$ and $q$ are replaced by $\hat{p}$ and $\hat{q}$. Consequently, for a fixed $r > 0$, from (2.3), we require $p \geq p - q \gtrsim \mathcal{O}(\frac{1}{\sqrt{n}})$ for success. However, setting the unobserved entries to 0, yields $\mathbf{A}_{i,j} = 0$ with probability $1 - rp$ inside the clusters and $1 - rq$ outside the clusters. This is equivalent to a fully observed graph where $p$ and $q$ are replaced by $rp$ and $rq$. In this case, we can allow $p \approx \mathcal{O}(\frac{1}{n})$ for success which is order-wise better, and matches the results in McSherry [27]. Intuitively, clustering a fully observed graph with parameters $\hat{p} = rp + 1 - r$ and $\hat{q} = rq + 1 - r$ is much more difficult than one with $rp$ and $rq$, since the links are *more noisy* in the former case. Hence, while it is beneficial to leave the unobserved entries blank in Program 1.1, for Program 1.4 it is in fact beneficial to set the unobserved entries to 0.

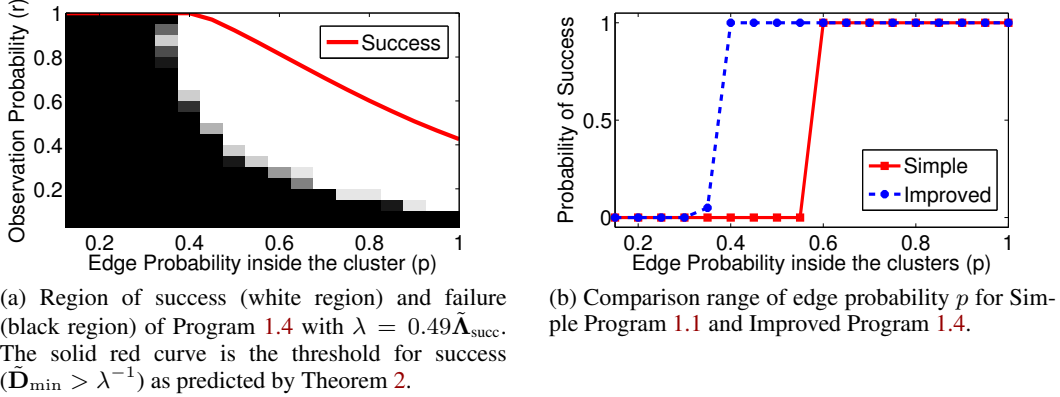

(a) Region of success (white region) and failure (black region) of Program 1.4 with $\lambda = 0.49\tilde{\mathbf{\Lambda}}_{\text{succ}}$. The solid red curve is the threshold for success $(\tilde{\mathbf{D}}_{\min} > \lambda^{-1})$ as predicted by Theorem 2.

(b) Comparison range of edge probability $p$ for Simple Program 1.1 and Improved Program 1.4.

Figure 2: Simulation results for Improved Program.

## 3 Experimental Results

We implement Program 1.1 and 1.4 using the inexact augmented Lagrange method of multipliers [28]. Note that this method solves the Program 1.1 and 1.4 approximately. Further, the numerical imprecisions will prevent the entries of the output of the algorithms from being strictly equal to 0 or 1. We use the mean of all the entries of the output as a hard threshold to round each entry. That is, if an entry is less than the threshold, it is rounded to 0 and to 1 otherwise. We compare the output of the algorithm after rounding to the optimal solution $(\mathbf{L}^0)$, and declare success if the number of wrong entries is less than $0.1\%$.

**Set Up:** We consider at an unweighted graph on $n = 600$ nodes with 3 disjoint clusters. For simplicity the clusters are of equal size $n_1 = n_2 = n_3$, and the edge probability inside the clusters are same $p_1 = p_2 = p_3 = p$. The edge probability outside the clusters is fixed, $q = 0.1$. We generate the adjacency matrix randomly according to the Stochastic Block Model 2.1 and Partial Observation Model 2.2. All the results are an average over 20 experiments.

### 3.1 Simulations for Simple Convex Program

**Dependence between $r$ and $p$:** In the first set of experiments we keep $n_1 = n_2 = n_3 = 200$, and vary $p$ from $0.55$ to 1 and $r$ from $0.05$ to 1 in steps of $0.05$.

**Dependence between $n_{\min}$ and $r$:** In the second set of experiments we keep the edge probability inside the clusters fixed, $p = 0.85$. The cluster size is varied from $n_{\min} = 20$ to $n_{\min} = 200$ in steps of 20 and $r$ is varied from $0.05$ to 1 in steps of $0.05$.

In both the experiments, we set the regularization parameter $\lambda = 1.01\mathbf{D}_{\min}^{-1}$, ensuring that $\mathbf{D}_{\min} > 1/\lambda$, enabling us to focus on observing the transition around $\mathbf{\Lambda}_{\text{succ}}$ and $\mathbf{\Lambda}_{\text{fail}}$. The outcome of the experiments are shown in the Figures 1a and 1b. The experimental region of success is shown in white and the region of failure is shown in black. The theoretical region of success is about the solid red curve $(\lambda < \mathbf{\Lambda}_{\text{succ}})$ and the region of failure is below dashed green curve $(\lambda > \mathbf{\Lambda}_{\text{fail}})$. As we can see the transition indeed occurs between the two thresholds $\mathbf{\Lambda}_{\text{succ}}$ and $\mathbf{\Lambda}_{\text{fail}}$.

### 3.2 Simulations for Improved Convex Program

We keep the cluster size, $n_1 = n_2 = n_3 = 200$ and vary $p$ from $0.15$ to 1 and $r$ from $0.05$ to 1 in steps of $0.05$. We set the regularization parameter, $\lambda = 0.49\tilde{\mathbf{\Lambda}}_{\text{succ}}$, ensuring that $\lambda < \tilde{\mathbf{\Lambda}}_{\text{succ}}$, enabling us to focus on observing the condition of success around $\tilde{\mathbf{D}}_{\min}$. The outcome of this experiment is shown in the Figure 2a. The experimental region of success is shown in white and region of failure is shown in black. The theoretical region of success is above solid red curve.

**Comparison with the Simple Convex Program:** In this experiment, we are interested in observing the range of $p$ for which the Programs 1.1 and 1.4 work. Keeping the cluster size $n_1 = n_2 = n_3 =$

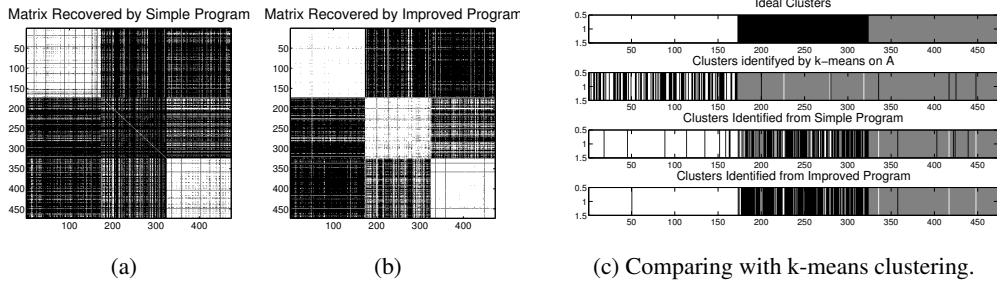

(a)                  (b)            (c) Comparing with k-means clustering.

Figure 3: Result of using (a) Program 1.1 (Simple) and (b) Program 1.4 (Improved) on the real data set. (c) Comparing the clustering output after running Program 1.1 and Program 1.4 with the output of applying k-means clustering directly on $A$ (with unknown entries set to 0).

200 and $r = 1$, we vary the edge probability inside the clusters from $p = 0.15$ to $p = 1$ in steps of 0.05. For each instance of the adjacency matrix, we run both Program 1.1 and 1.4. We plot the probability of success of both the algorithms in Figure 2b. As we can observe, Program 1.1 starts succeeding only after $p > 1/2$, whereas for Program 1.4 it starts at $p \approx 0.35$.

### 3.3 Labeling Images: Amazon MTurk Experiment

Creating a training dataset by labeling images is a tedious task. It would be useful to crowdsource this task instead. Consider a specific example of a set of images of dogs of different breeds. We want to cluster them such that the images of dogs of the same breed are in the same cluster. One could show a set of images to each worker, and ask him/her to identify the breed of dog in each of those images. But such a task would require the workers to be experts in identifying the dog breeds. A relatively reasonable task is to ask the workers to compare pairs of images, and for each pair, answer whether they think the dogs in the images are of the same breed or not. If we have $n$ images, then there are $\binom{n}{2}$ distinct pairs of images, and it will pretty quickly become unreasonable to compare all possible pairs. This is an example where we could obtain a subset of the data and try to cluster the images based on the partial observations.

**Image Data Set:** We used images of 3 different breeds of dogs : Norfolk Terrier (172 images), Toy Poodle (151 images) and Bouvier des Flandres (150 images) from the Standford Dogs Dataset [29]. We uploaded all the 473 images of dogs on an image hosting server (we used imgur.com).

**MTurk Task:** We used Amazon Mechanical Turk [30] as the platform for crowdsourcing. For each worker, we showed 30 pairs of images chosen randomly from the $\binom{n}{2}$ possible pairs. The task assigned to the worker was to compare each pair of images, and answer whether they think the dogs belong to the same breed or not. If the worker's response is a "yes", then there we fill the entry of the adjacency matrix corresponding to the pair as 1, and 0 if the answer is a "no".

**Collected Data:** We recorded around 608 responses. We were able to fill $16,750$ out of $111,628$ entries in $\mathbf{A}$. That is, we observed $15\%$ of the total number of entries. Compared with true answers (which we know a priori), the answers given by the workers had around $23.53\%$ errors (3941 out of 16750). The empirical parameters for the partially observed graph thus obtained is shown Table 1.

We ran Program 1.1 and Program 1.4 with regularization parameter, $\lambda = 1/\sqrt{n}$. Further, for Program 1.4, we set the size of the cluster region, $\mathcal{R}$ to 0.125 times $\binom{n}{2}$. Figure 3a shows the recovered matrices. Entries with value 1 are depicted by white and 0 is depicted by black. In Figure 3c we compare the clusters output by running the k-means algorithm directly on the adjacency matrix $\mathbf{A}$ (with unknown entries set to 0) to that obtained by running k-means algorithm on the matrices recovered after running Program 1.1 (Simple Program) and Program 1.4 (Improved Program) respectively. The overall error with k-means was $40.8\%$ whereas the error significantly reduced to $15.86\%$ and $7.19\%$ respectively when we used the matrices recoverd from Programs 1.1 and 1.4 respectively (see Table 2). Further, note that for running the k-means algorithm we need to know the exact number of clusters. A common heuristic is to identify the top $K$ eigenvalues that are much

Table 1: Empirical Parameters from the real data.

| Params | Value | Params | Value |
|---|---|---|---|
| $n$ | 473 | $r$ | 0.1500 |
| $K$ | 3 | $q$ | 0.1929 |
| $n_1$ | 172 | $p_1$ | 0.7587 |
| $n_2$ | 151 | $p_2$ | 0.6444 |
| $n_3$ | 150 | $p_3$ | 0.7687 |

Table 2: Number of miss-classified images

| Clusters→ | 1 | 2 | 3 | Total |
|---|---|---|---|---|
| K-means | 39 | 150 | 4 | 193 |
| Simple | 9 | 57 | 8 | 74 |
| Improved | 1 | 29 | 4 | 34 |

larger than the rest. In Figure 4 we plot the sorted eigenvalues for the adjacency matrix **A** and the recovered matrices. We can see that the top 3 eigen values are very easily distinguished from the rest for the matrix recovered after running Program 1.4.

A sample of the data is shown in Figure 5. We observe that factors such as color, grooming, posture, face visibility etc. can result in confusion while comparing image pairs. Also, note that the ability of the workers to distinguish the dog breeds is neither guaranteed nor uniform. Thus, the edge probability inside and outside clusters are not uniform. Nonetheless, Programs 1.1 and Program 1.4, especially Program 1.4, are quite successful in clustering the data with only 15% observations.

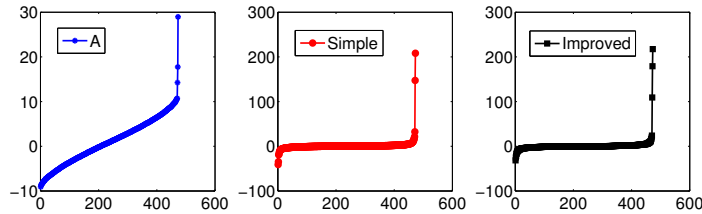

Figure 4: Plot of sorted eigen values for (1) Adjacency matrix with unknown entries filled by 0, (2) Recovered adjacency matrix from Program 1.1, (3) Recovered adjacency matrix from Program 1.4

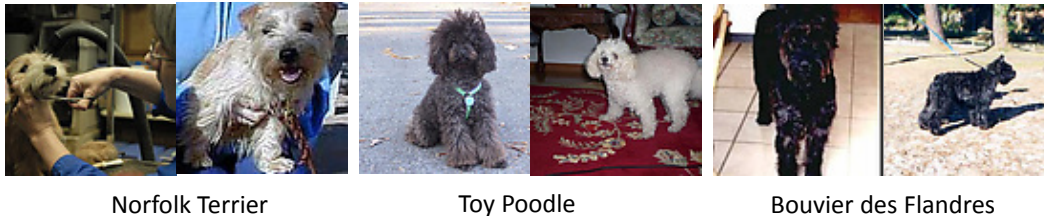

| Norfolk Terrier | Toy Poodle | Bouvier des Flandres |
|---|---|---|

Figure 5: Sample images of three breeds of dogs that were used in the MTurk experiment.

The authors thank the anonymous reviewers for their insightful comments. This work was supported in part by the National Science Foundation under grants CCF-0729203, CNS-0932428 and CIF-1018927, by the Office of Naval Research under the MURI grant N00014-08-1-0747, and by a grant from Qualcomm Inc. The first author is also supported by the Schlumberger Foundation Faculty for the Future Program Grant.

## Footnotes

[1]The proofs for Theorems 1 and 2 are provided in the supplementary material.

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
