[Supplementary Material]

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

_{1 \leq i \leq K} n_i$, $p_{min} := \min_{1 \leq i \leq K} p_i$ and $n_{\max} := \max_{1 \leq i \leq K} n_i$.

The following definitions are important to describe our results.

- Define $\mathbf{D}_i := n_i\, r\, (2p_i - 1)$ as the **effective density** of cluster $i$ and $\mathbf{D}_{\min} = \min_{1 \leq i \leq K} \mathbf{D}_i$.

- $\gamma_{\mathrm{succ}} := \max_{1 \leq i \leq K} 2r\sqrt{n_i}\sqrt{2(\frac{1}{r} - 1) + 4\left(q(1-q) + p_i(1-p_i)\right)}$ and $\gamma_{\mathrm{fail}} := \sum_{i=1}^{K} \frac{n_i^2}{n}$

- $\mathbf{\Lambda}_{\mathrm{succ}}^{-1} := 2r\sqrt{n}\sqrt{\frac{1}{r} - 1 + 4q(1-q)} + \gamma_{\mathrm{succ}}$ and $\mathbf{\Lambda}_{\mathrm{fail}}^{-1} := \sqrt{rq(n - \gamma_{\mathrm{fail}})}$.

We note that the thresholds, $\mathbf{\Lambda}_{\mathrm{succ}}$ and $\mathbf{\Lambda}_{\mathrm{fail}}$ depend only the parameters of the model. Some simple algebra shows that $\mathbf{\Lambda}_{\mathrm{succ}} < \mathbf{\Lambda}_{\mathrm{fail}}$.

**Theorem 1** (Simple Program). *Consider a random graph generated according to the Partial Observation Model of Definition (2.2) with K disjoint clusters of sizes $\{n_i\}_{i=1}^{K}$, and probabilities $\{p_i\}_{i=1}^{K}$ and q, such that $p_{min} > \frac{1}{2} > q > 0$. Given $\epsilon > 0$, there exists positive constants $c_1', c_2'$ such that,*

1. *If $\lambda \geq (1 + \epsilon)\mathbf{\Lambda}_{fail}$, then Program 1.1 fails to correctly recover the clusters with probability $1 - c_1' \exp(-c_2'|\mathcal{R}^c|)$.*

2. *If $0 < \lambda \leq (1 - \epsilon)\mathbf{\Lambda}_{succ}$,*

   - *If $\mathbf{D}_{\min} \geq (1 + \epsilon)\frac{1}{\lambda}$, then Program 1.1 succeeds in correctly recovering the clusters with probability $1 - c_1' n^2 \exp(-c_2' n_{\min})$.*
   - *If $\mathbf{D}_{\min} \leq (1 - \epsilon)\frac{1}{\lambda}$, then Program 1.1 fails to correctly recover the clusters with probability $1 - c_1' \exp(-c_2' n_{\min})$.*

**Discussion:**

1. Theorem 1 characterizes the success and failure of Program 1.1 as a function of the regularization parameter $\lambda$. In particular, if $\lambda > \mathbf{\Lambda}_{\mathrm{fail}}$, Program 1.1 fails with high probability. If $\lambda < \mathbf{\Lambda}_{\mathrm{succ}}$, Program 1.1 succeeds with high probability *if and only if* $\mathbf{D}_{\min} > \frac{1}{\lambda}$. However, Theorem 1 has nothing to say about $\mathbf{\Lambda}_{\mathrm{succ}} < \lambda < \mathbf{\Lambda}_{\mathrm{fail}}$.

2. **Small Cluster Regime:** When $n_{\max} = o(n)$, we have $\mathbf{\Lambda}_{\mathrm{succ}}^{-1} = 2r\sqrt{n}\sqrt{\left(\frac{1}{r} - 1 + 4q(1-q)\right)}$. For simplicity let $p_i = p, \forall\, i$, which yields $\mathbf{D}_{\min} = n_{\min} r(2p-1)$. Then $\mathbf{D}_{\min} > \mathbf{\Lambda}_{\mathrm{succ}}^{-1}$ implies,

$$ n_{\min} > \frac{2\sqrt{n}}{2p - 1}\sqrt{\left(\frac{1}{r} - 1 + 4q(1-q)\right)}, \tag{2.2} $$

   giving a lower bound on the minimum cluster size that is sufficient for success.

## 2.3 Results: Improved Convex Program

The following definitions are critical to describe our results.

- Define $\tilde{\mathbf{D}}_i := n_i\, r\, (p_i - q)$ as the effective density of cluster $i$ and $\tilde{\mathbf{D}}_{\min} = \min_{1 \leq i \leq K} \tilde{\mathbf{D}}_i$.

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

# 4 Proof of Results for Simple Convex Program

Let $1 \geq p_{min} > \frac{1}{2} > q > 0$ and $0 \leq r \leq 1$. $\mathcal{G}$ be a random graph generated according to the stochastic block model 2.1 with cluster sizes $\{n_i\}_{i=1}^K$. Let the observation model be as defined in (Defn 2.2). Theorem 1 is based on the following lemmas:

**Lemma 4.1.** *If $\lambda > \Lambda_{fail}$, then $(\mathbf{L}^0, \mathbf{S}^0)$ is not an optimal solution to the Program 1.1 with high probability.*

**Lemma 4.2.** *If $\lambda < \Lambda_{succ}$ and $\mathbf{D}_{\min} > \frac{1}{\lambda}$, then $(\mathbf{L}^0, \mathbf{S}^0)$ is the unique optimal solution to Program 1.1 with high probability.*

Before we proceed, we need some additional notations. Let $\mathcal{R}_{i,j} = \mathcal{C}_i \times \mathcal{C}_j$ for $1 \leq i, j \leq K + 1$. One can see that $\{\mathcal{R}_{i,j}\}$ divides $[n] \times [n]$ into $(K+1)^2$ disjoint regions similar to a grid which is illustrated in the Figure 6. Thus, $\mathcal{R}_{i,i}$ is the region induced by $i$'th cluster for any $1 \leq i \leq K$.

Let $\Gamma^{out}$ be the set of entries of adjacency matrix that are *not* observed. Let $\mathcal{A} \subseteq [n] \times [n]$ be the set of observed coordinates of $\mathbf{A_{obs}}$. Let $\mathcal{A}_1 \subseteq [n] \times [n]$ be the set of nonzero coordinates of $\mathbf{A_{obs}}$, and $\mathcal{A}_0 \subseteq [n] \times [n]$ be the set of coordinates of $\mathbf{A_{obs}}$ that are zero. Then the sets,

1. $\mathcal{A}_1 \cap \mathcal{R}$ corresponds to the edges inside the clusters that are observed.

2. $\mathcal{A}_1 \cap \mathcal{R}^c$ corresponds to the set of edges outside the clusters that are observed.

3. $\mathcal{A}_0 \cap \mathcal{R}$ corresponds to the missing edges inside the clusters, that are observed (that is, we know that the edge does not exist).

Let $c$ and $d$ be positive integers. Consider a matrix, $\mathbf{X} \in \mathbb{R}^{c \times d}$. Let $\beta$ be a subset of $[c] \times [d]$. Then, let $\mathbf{X}_\beta$ denote the matrix induced by the entries of $\mathbf{X}$ on $\beta$ i.e.,

$$(\mathbf{X}_\beta)_{i,j} = \begin{cases} \mathbf{X}_{i,j} & \text{if } (i,j) \in \beta \\ 0 & \text{otherwise .} \end{cases}$$

In other words, $\mathbf{X}_\beta$ is a matrix whose entries match those of $\mathbf{X}$ in the positions $(i,j) \in \beta$ and zero otherwise. For example, $\mathbb{1}_{\mathcal{A}^{obs}}^{n \times n} = \mathbf{A}^{obs}$. Given a matrix $\mathbf{X}$, $\text{sum}(\mathbf{X})$ will denote the sum of all entries of $\mathbf{X}$. Finally, we introduce the following parameter which will be useful for the subsequent analysis. Given $q, \{p_i\}_{i=1}^K$, let,

$$\mathbf{D}_{\mathcal{A}} = \frac{1}{2} \min \left\{ r(1-2q), \left\{ r(2p_i - 1) - \frac{1}{\lambda n_i} \right\}_{i=1}^K \right\} \tag{4.1}$$

$$= \frac{1}{2} \min \left\{ r(1-2q), \frac{\mathbf{D}_i - \lambda^{-1}}{n_i} \right\}$$

For our proofs, we will make use of the following Big O notation. $f(n) = \Omega(n)$ will mean there exists a positive constant $c$ such that for sufficiently large $n$, $f(n) \geq cn$. $f(n) = O(n)$ will mean there exists a positive constant $c$ such that for sufficiently large $n$, $f(n) \leq cn$.

## 4.1 Proof of Lemma 4.1

Lagrange for the problem (1.1) can be written as follows

$$\mathscr{L}(\mathbf{L}, \mathbf{S}; \mathbf{M}, \mathbf{N}) \quad = \quad \|\mathbf{L}\|_\star + \lambda \|\mathbf{S_{obs}}\|_1 + \mathrm{trace}(\mathbf{M}(\mathbf{L} - \mathbb{1}\mathbb{1}^T)) - \mathrm{trace}(\mathbf{N}\mathbf{L}). \qquad (4.2)$$

where $\mathbf{M}$ and $\mathbf{N}$ are dual variables corresponding to the inequality constraints (1.2).

For $\mathbf{L}^0$ to be an optimal solution to (1.1), it has to satisfy the KKT conditions. Therefore, the subgradient of (4.2) at $\mathbf{L}^0$ has to be 0, i.e.,

$$\partial\|\mathbf{L}^0\|_\star + \lambda\,\partial\|\mathbf{A_{obs}} - \mathbf{L_{obs}}^0\|_1 + \mathbf{M}^0 - \mathbf{N}^0 = 0. \qquad (4.3)$$

where $\mathbf{M}^0$ and $\mathbf{N}^0$ are optimal dual variables, and $\partial\|\mathbf{L}^0\|_\star$ and $\partial\|\mathbf{S}^0\|_1$ are subgradients of nuclear norm and $\ell_1$-norm respectively at the points $(\mathbf{L}^0, \mathbf{S}^0)$. Note that in the standard notation, $\partial\|\mathbf{x}\|_\star$ denotes the set of all subgradients, i.e., the subdifferential. We have slightly abused the notation by denoting a subgradient of a norm $\|\cdot\|_\star$ at the point $\mathbf{x}$ by $\partial\|\mathbf{x}\|_\star$.

Also, by complementary slackness,

$$\mathrm{trace}(\mathbf{M}^0(\mathbf{L}^0 - \mathbb{1}\mathbb{1}^T)) = 0, \qquad (4.4)$$

and

$$\mathrm{trace}(\mathbf{N}^0\mathbf{L}^0) = 0. \qquad (4.5)$$

From (5.1) and (4.4), (4.5), we have $(\mathbf{M}^0)_{\mathcal{R}} \geq 0$, $(\mathbf{M}^0)_{\mathcal{R}^c} = 0$, $(\mathbf{N}^0)_{\mathcal{R}} = 0$ and $(\mathbf{N}^0)_{\mathcal{R}^c} \geq 0$. Hence $(\mathbf{M}^0 - \mathbf{N}^0)_{\mathcal{R}} \geq 0$ and $(\mathbf{M}^0 - \mathbf{N}^0)_{\mathcal{R}^c} \leq 0$.

Let $\mathbf{L}^0 = \mathbf{U}\Lambda\mathbf{U}^T$, where $\Lambda = \mathrm{diag}\{n_1, n_2, \ldots, n_K\}\mathbf{U} = [\mathbf{u}_1 \;\ldots\; \mathbf{u}_K] \in \mathbb{R}^{n \times K}$,

$$\mathbf{u}_{l,i} = \begin{cases} \frac{1}{\sqrt{n_l}} & \text{if } i \in \mathcal{C}_l \\ 0 & \text{else.} \end{cases} \qquad (4.6)$$

Then the subgradient $\partial\|\mathbf{L}^0\|_\star$ is of the form $\mathbf{U}\mathbf{U}^T + \mathbf{W}$ such that $\mathbf{W} \in \{\mathbf{X} : \mathbf{X}\mathbf{U} = \mathbf{U}^T\mathbf{X} = 0, \|\mathbf{X}\| \leq 1\}$. The subgradient $\partial\|\mathbf{S}^0\|_1$ is of the form $\mathrm{sign}(\mathbf{S}^0) + \mathbf{Q}$ where $\mathbf{Q}_{i,j} = 0$ if $\mathbf{S}_{i,j} \neq 0$ and $\|\mathbf{Q}\|_\infty \leq 1$. Further, note that the subgradient of $\mathbf{S}^0$ over unobserved entries is zero. That is, $\partial\|\mathbf{S}^0_{unobs}\|_1 = 0$ since $\mathbf{S}^0_{unobs} = 0$.

From (4.3), we have

$$\mathbf{U}\mathbf{U}^T + \mathbf{W} - \lambda\left(\mathrm{sign}(\mathbf{S}^0) + \mathbf{Q}\right) + (\mathbf{M}^0 - \mathbf{N}^0) = 0. \qquad (4.7)$$

Consider the sum of the entires corresponding to the cluster $i$ ($\mathcal{R}_{i,i}$), i.e.,

$$\underbrace{\mathrm{sum}\,(\mathbf{L}^0)_{\mathcal{R}_{i,i}}}_{n_i} - \mathrm{sum}\left(\lambda\left(\mathrm{sign}(\mathbf{S}^0) + \mathbf{Q}\right)_{\mathcal{R}_{i,i}}\right)$$

$$+ \underbrace{\mathrm{sum}\,(\mathbf{M}^0 - \mathbf{N}^0)_{\mathcal{R}_{i,i}}}_{\geq 0} = 0 \qquad (4.8)$$

Since each entry of the adjacency matrix is observed with probability $r$, and the probability of missing edge inside cluster $i$ is $1 - p_i$, we note that $(\mathbf{S}^0_{\mathcal{R}_{i,i}})_{l,m} \neq 0$ with probability $r(1 - p_i)$. Recall that $\mathbf{Q}_{l,m} = 0$ if $\mathbf{S}^0_{l,m} \neq 0$.

Then by Bernstein's inequality and using $\|\mathbf{Q}\|_\infty \leq 1$, with probability $1 - \exp\left(-\Omega(n_i^2)\right)$ we have $\mathrm{sum}\left(\mathrm{sign}(\mathbf{S}^0)\right) = -n_i^2 r(1 - p_i)$ and $\mathrm{sum}\,(\mathbf{Q}) \leq n_i^2 r p_i$ .

Thus,

$$-\mathrm{sum}\left(\lambda\left(\mathrm{sign}(\mathbf{S}^0) + \mathbf{Q}\right)_{\mathcal{R}_{i,i}}\right) \quad \geq \quad \lambda n_i^2 r(1 - p_i) - \lambda n_i^2 r p_i$$

$$= \quad \lambda n_i^2 r\left(1 - 2p_i\right). \qquad (4.9)$$

and hence LHS of equation (4.8) can be lower bounded as ,

$$n_i - \text{sum}\left(\lambda\left(\text{sign}(\mathbf{S}^0) + \mathbf{Q}\right)_{\mathcal{R}_{i,i}}\right) + \underbrace{\text{sum}\left(\mathbf{M}^0 - \mathbf{N}^0\right)_{\mathcal{R}_{i,i}}}_{\geq 0} \geq n_i + \lambda n_i^2 r\left(1 - 2p_i\right). \quad (4.10)$$

We see that $n_i r\left(2p_i - 1\right) < \frac{1}{\lambda}$ would imply $n_i + \lambda n_i^2 r\left(1 - 2p_i\right) > 0$, in which case, the equation (4.3) does not hold. Hence $\mathbf{L}^0$ cannot be an optimal solution to the Program 1.1. (Note that, $p_i > \frac{1}{2}$ and hence $2p_i - 1 > 0$.)

Notice that $\left(\mathbf{U}\mathbf{U}^T\right)_{\mathcal{R}^c} = 0$ and the entries of $-\left(\text{sign}(\mathbf{S}^0) + \mathbf{Q}\right)$ and $\mathbf{M}^0 - \mathbf{N}^0$ over $\mathcal{R}^c \cap \mathcal{A}_1$ are negative. Hence from the equation (4.7),

$$\|\mathbf{W}\|_F^2 \quad \geq \quad \|\left(\mathbf{U}\mathbf{U}^T + \mathbf{W}\right)_{(\mathcal{R}^c \cap \mathcal{A}_1)}\|_F^2 \geq \|\lambda\left(\text{sign}(\mathbf{S}^0) + \mathbf{Q}\right)_{(\mathcal{R}^c \cap \mathcal{A}_1)}\|_F^2. \quad (4.11)$$

Recall that $\mathbf{S}^0_{(\mathcal{R}_c \cap \mathcal{A}_1)} \neq 0$ and hence $\mathbf{Q}_{(\mathcal{R}_c \cap \mathcal{A}_1)} = 0$. Further, recall that by the Stochastic Block Model, each entry of $\mathbf{A}$ over $\mathcal{R}^c$ is non-zero with probability $q$ and by observation model (Defn 2.2), each entry of $\mathbf{A}$ is observed with probability $r$. Hence with probability at least $1 - \exp\left(-\Omega(|\mathcal{R}_c|)\right)$, $|\mathcal{R}_c \cap \mathcal{A}_1| = rq(n^2 - \sum_{i=1}^{K} n_i^2)$. Thus from equation (4.11) we have,

$$\|\mathbf{W}\|_F^2 \geq \lambda^2 rq(n^2 - \sum_{i=1}^{K} n_i^2), \quad (4.12)$$

Recall that $\|\mathbf{W}\| \leq 1$ should hold true for $\left(\mathbf{L}^0, \mathbf{S}^0\right)$ to be an optimal solution to Program 1.1. $\|\mathbf{W}\| = |\sigma_{\max}(\mathbf{W})| \geq \frac{\|\mathbf{W}\|_F}{\sqrt{n}}$, which on combining with equation (4.12) gives us,

$$\|\mathbf{W}\| \geq \lambda\sqrt{\frac{rq\left(n^2 - \sum_{i=1}^{K} n_i^2\right)}{n}}.$$

So, if $\lambda\sqrt{rq\left(n^2 - \sum_{i=1}^{K} n_i^2\right)/n} > 1$ then, $\left(\mathbf{L}^0, \mathbf{S}^0\right)$ cannot be an optimal solution to Program 1.1. This gives us the result in Lemma 4.1.

## 4.2 Proof of Lemma 4.2

In order to show that $\left(\mathbf{L}^0, \mathbf{S}^0\right)$ is the unique optimal solution to the Program 1.1, we need to prove that for all feasible perturbations $(\mathbf{E}^L, \mathbf{E}^S)$,

$$(\|\mathbf{L}^0 + \mathbf{E}^L\|_\star + \lambda\|\mathbf{S}^0 + \mathbf{E}^S\|_1) - (\|\mathbf{L}^0\|_\star + \lambda\|\mathbf{S}^0\|_1) > 0. \quad (4.13)$$

We note that $\mathbf{S}$ can be split as $\mathbf{S} = \mathbf{S}_{\mathbf{obs}} + \mathbf{S}_{\mathbf{rest}}$, where $\mathbf{S}_{\mathbf{rest}}$ denotes the entries of $\mathbf{S}$ other than those corresponding to the observed entries of $\mathbf{A}$. Furthermore, we claim that at the optimal, $\mathbf{S}_{\mathbf{rest}} = 0$, since if otherwise, the objective can be strictly decreased by setting $\mathbf{S}_{\mathbf{rest}} = 0$. Hence, $\mathbf{S} = \mathbf{S}_{\mathbf{obs}}$.

We can lower bound the LHS of the equation (4.13) using the subgradients as follows,

$$(\|\mathbf{L}^0 + \mathbf{E}^L\|_\star + \lambda\|\mathbf{S}^0 + \mathbf{E}^S\|_1) - (\|\mathbf{L}^0\|_\star + \lambda\|\mathbf{S}^0\|_1) \geq \langle\partial\|\mathbf{L}^0\|_\star, \mathbf{E}^L\rangle + \lambda\langle\partial\|\mathbf{S}^0\|_1, \mathbf{E}^S\rangle, \quad (4.14)$$

where $\partial\|\mathbf{L}^0\|_\star$ and $\partial\|\mathbf{S}^0\|_1$ are subgradients of nuclear norm and $\ell_1$-norm respectively at the points $\left(\mathbf{L}^0, \mathbf{S}^0\right)$. Note that in the standard notation, $\partial\|\mathbf{x}\|_*$ denotes the set of all subgradients, i.e., the subdifferential. We have slightly abused the notation by denoting a subgradient of a norm $\|\cdot\|_*$ at the point $\mathbf{x}$ by $\partial\|\mathbf{x}\|_*$.

To make use of (4.14), it is very important to choose good subgradients. In the following section we will focus on construction of such subgradients.

#### 4.2.1 Subgradient construction

Recall that, $\mathbf{L}^0 = \mathbf{U}\Lambda\mathbf{U}^T$, where $\Lambda = \mathrm{diag}\{n_1, n_2, \ldots, n_K\}$ and $\mathbf{U} = [\mathbf{u}_1 \ \ldots \ \mathbf{u}_K] \in \mathbb{R}^{n \times K}$, with $\mathbf{u}_l$ as defined before. Then the subgradient $\partial\|\mathbf{L}^0\|_\star$ is of the form $\mathbf{U}\mathbf{U}^T + \mathbf{W}$ such that $\mathbf{W} \in \mathcal{M}_U := \{\mathbf{X} : \mathbf{X}\mathbf{U} = \mathbf{U}^T\mathbf{X} = 0, \|\mathbf{X}\| \leq 1\}$. $\|.\|$ is spectral norm (maximum singular value). The subgradient $\partial\|\mathbf{S}^0\|_1$ is of the form $\mathrm{sign}(\mathbf{S}^0) + \mathbf{Q}$ where $\mathbf{Q}_{i,j} = 0$ if $\mathbf{S}^0_{i,j} \neq 0$ and $\|\mathbf{Q}\|_\infty \leq 1$.

$$\|\mathbf{L}^0 + \mathbf{E}^L\|_\star + \lambda\,\|\mathbf{S}^0 + \mathbf{E}^S\|_1 - (\|\mathbf{L}^0\|_\star + \lambda\,\|\mathbf{S}^0\|_1) \geq \langle\partial\|\mathbf{L}^0\|_\star, \mathbf{E}^L\rangle + \lambda\langle\partial\|\mathbf{S}^0\|_1, \mathbf{E}^S\rangle$$
$$= \langle\mathbf{U}\mathbf{U}^T + \mathbf{W}, \mathbf{E}^L\rangle + \lambda\langle\mathrm{sign}(\mathbf{S}^0) + \mathbf{Q}, \mathbf{E}^S\rangle$$

Note that, due to the condition $\mathbf{L_{obs}} + \mathbf{S_{obs}} = \mathbf{A_{obs}}$, we have $\mathbf{E}^S = \mathbf{E_{obs}}^L$. Further, note that $\mathrm{sign}(\mathbf{S}^0) = \mathbb{1}^{n \times n}_{\mathcal{A}_1 \cap \mathcal{R}^c} - \mathbb{1}^{n \times n}_{\mathcal{A}_0 \cap \mathcal{R}}$. Choosing $\mathbf{Q} = \mathbb{1}^{n \times n}_{\mathcal{A}_1 \cap \mathcal{R}} - \mathbb{1}^{n \times n}_{\mathcal{A}_0 \cap \mathcal{R}^c}$, we get,

$$\|\mathbf{L}^0 + \mathbf{E}^L\|_\star + \lambda\,\|\mathbf{S}^0 + \mathbf{E}^S\|_1 - (\|\mathbf{L}^0\|_\star + \lambda\,\|\mathbf{S}^0\|_1) \geq \langle\mathbf{W}, \mathbf{E}^L\rangle$$
$$+ \underbrace{\sum_{i=1}^{K} \frac{1}{n_i}\mathrm{sum}(\mathbf{E}_{R_{i,i}}) + \lambda\left(\mathrm{sum}(\mathbf{E}^L_{\mathcal{A}_0}) - \mathrm{sum}(\mathbf{E}^L_{\mathcal{A}_1})\right)}_{:=g(\mathbf{E}^L)}$$

$$(4.15)$$

From this point onward, for simplicity we will ignore the superscript L on $\mathbf{E}^L$ and just use $\mathbf{E}$.

Define,

$$g(\mathbf{E}) := \sum_{i=1}^{K} \frac{1}{n_i}\mathrm{sum}(\mathbf{E}_{\mathcal{R}_{i,i}}) + \lambda\left(\mathrm{sum}(\mathbf{E}_{\mathcal{A}_0}) - \mathrm{sum}(\mathbf{E}_{\mathcal{A}_1})\right). \qquad (4.16)$$

Also, define $f(\mathbf{E}, \mathbf{W}) := g(\mathbf{E}) + \langle\mathbf{W}, \mathbf{E}\rangle$. Our aim is to show that for all feasible perturbations $\mathbf{E}$, there exists $\mathbf{W}$ such that,

$$f(\mathbf{E}, \mathbf{W}) = g(\mathbf{E}) + \langle\mathbf{W}, \mathbf{E}\rangle > 0. \qquad (4.17)$$

Note that $g(\mathbf{E})$ does not depend on $\mathbf{W}$.

**Lemma 4.3.** *Given* $\mathbf{E}$*, assume there exists* $\mathbf{W} \in \mathcal{M}_{\mathbf{U}}$ *with* $\|\mathbf{W}\| < 1$ *such that* $f(\mathbf{E}, \mathbf{W}) \geq 0$*. Then at least one of the followings holds:*

- *There exists* $\mathbf{W}^* \in \mathcal{M}_{\mathbf{U}}$ *with* $\|\mathbf{W}^*\| \leq 1$ *and* $f(\mathbf{E}, \mathbf{W}^*) > 0$.

- *For all* $\mathbf{W} \in \mathcal{M}_{\mathbf{U}}$, $\langle\mathbf{E}, \mathbf{W}\rangle = 0$.

*Proof.* Let $c = 1 - \|\mathbf{W}\|$. Assume $\langle\mathbf{E}, \mathbf{W}'\rangle \neq 0$ for some $\mathbf{W}' \in \mathcal{M}_{\mathbf{U}}$. If $\langle\mathbf{E}, \mathbf{W}'\rangle > 0$, choose $\mathbf{W}^* = \mathbf{W} + c\mathbf{W}'$. Otherwise, choose $\mathbf{W}^* = \mathbf{W} - c\mathbf{W}'$. Since $\|\mathbf{W}'\| \leq 1$, we have, $\|\mathbf{W}^*\| \leq 1$ and $\mathbf{W}^* \in \mathcal{M}_{\mathbf{U}}$. Consequently,

$$f(\mathbf{E}, \mathbf{W}^*) = f(\mathbf{E}, \mathbf{W}) + \langle\mathbf{E}, c\mathbf{W}'\rangle > f(\mathbf{E}, \mathbf{W}) \geq 0 \qquad (4.18)$$

∎

Notice that, for all $\mathbf{W} \in \mathcal{M}_{\mathbf{U}}$, $\langle\mathbf{E}, \mathbf{W}\rangle = 0$ is equivalent to $\mathbf{E} \in \mathcal{M}_{\mathbf{U}}^\perp$ which is the orthogonal complement of $\mathcal{M}_{\mathbf{U}}$ in $\mathbb{R}^{n \times n}$. $\mathcal{M}_{\mathbf{U}}^\perp$ has the following characterization:

$$\mathcal{M}_{\mathbf{U}}^\perp = \{\mathbf{X} \in \mathbb{R}^{n \times n} : \mathbf{X} = \mathbf{U}\mathbf{M}^T + \mathbf{N}\mathbf{U}^T \text{ for some } \mathbf{M}, \mathbf{N} \in \mathbb{R}^{n \times K}\}. \qquad (4.19)$$

Now we have broken down our aim into two steps.

1. Construct $\mathbf{W} \in \mathcal{M}_{\mathbf{U}}$ with $\|\mathbf{W}\| < 1$, such that $f(\mathbf{E}, \mathbf{W}) \geq 0$ for all feasible perturbations $\mathbf{E}$.
2. For all non-zero feasible $\mathbf{E} \in \mathcal{M}_{\mathbf{U}}^\perp$, show that $g(\mathbf{E}) > 0$.

As a first step, in Section 4.3, we will argue that, under certain conditions, there exists a $\mathbf{W} \in \mathcal{M}_{\mathbf{U}}$ with $\|\mathbf{W}\| < 1$ such that with high probability, $f(\mathbf{E}, \mathbf{W}) \geq 0$ for all feasible $\mathbf{E}$. This $\mathbf{W}$ is called the dual certificate. Secondly, in Section 4.4, we will show that, under certain conditions, for all $\mathbf{E} \in \mathcal{M}_{\mathbf{U}}^\perp$ with high probability, $g(\mathbf{E}) > 0$. Finally, combining these two arguments, and using Lemma 4.3 we will conclude that $(\mathbf{L}^0, \mathbf{S}^0)$ is the unique optimal with high probability.

## 4.3 Showing existence of the dual certificate

Recall that

$$f(\mathbf{E}, \mathbf{W}) = \sum_{i=1}^{K} \frac{1}{n_i} \mathrm{sum}(\mathbf{E}_{\mathcal{R}_{i,i}}) + \langle \mathbf{E}, \mathbf{W} \rangle + \lambda \left( \mathrm{sum}\left(\mathbf{E}_{\mathcal{A}_0}\right) - \mathrm{sum}\left(\mathbf{E}_{\mathcal{A}_1}\right) \right)$$

$\mathbf{W}$ will be constructed from the candidate $\mathbf{W}_0$, which is given as follows.

### 4.3.1 Candidate $\mathbf{W}_0$

Based on Program 1.1, we propose the following,

$$\mathbf{W}_0 = \sum_{i=1}^{K} c_i \mathbb{1}^{n \times n}_{\mathcal{R}_{i,i}} + c \mathbb{1}^{n \times n}_{\mathcal{R}^c} + \lambda \left( \mathbb{1}^{n \times n}_{\mathcal{A}_1} - \mathbb{1}^{n \times n}_{\mathcal{A}_0} \right), \qquad (4.20)$$

where $\{c_i\}_{i=1}^{K}, c$ are real numbers to be determined.

$$f(\mathbf{E}, \mathbf{W}^0) = \sum_{i=1}^{K} (\frac{1}{n_i} + c_i)\, \mathrm{sum}(\mathbf{E}_{\mathcal{R}_{i,i}}) + c\, \mathrm{sum}(\mathbf{E}_{\mathcal{R}^c})$$

Note that $\mathbf{W}_0$ is a random matrix where randomness is due to $\mathbf{A_{obs}}$. In order to ensure a small spectral norm, we will set its expectation to 0, i.e., we will choose $c, \{c_i\}'s$ to ensure that $\mathbb{E}[\mathbf{W}_0] = 0$.

Following from the partially observed Stochastic Block Model (Defn 2.1 and 2.2), the expectation of an entry of $\mathbf{W}_0$ on $\mathcal{R}_{i,i}$ (region corresponding to cluster $i$) and $\mathcal{R}^c$ (region outside the clusters) is $c_i + \lambda r(2p_i - 1)$ and $c + \lambda r(2q - 1)$ respectively. Hence, we set,

$$c_i = -\lambda r(2p_i - 1) \quad \text{and} \quad c = -\lambda r(2q - 1),$$

With these, choices, the candidate $\mathbf{W}_0$ and $f(\mathbf{E}, \mathbf{W}_0)$ take the following forms,

$$\mathbf{W}_0 = \lambda \left[ \sum_{i=1}^{K} (1 + r(1 - 2p_i))\, \mathbb{1}^{n \times n}_{\mathcal{R}_{i,i} \cap \mathcal{A}_1} + (-1 + r(1 - 2p_i))\, \mathbb{1}^{n \times n}_{\mathcal{R}_{i,i} \cap \mathcal{A}_0} + r(1 - 2p_i) \mathbb{1}^{n \times n}_{\mathcal{R}_{i,i} \cap \Gamma^{out}} \right]$$

$$+ \lambda \left[ (1 + r(1 - 2q))\, \mathbb{1}^{n \times n}_{\mathcal{R}^c \cap \mathcal{A}_1} + (-1 + r(1 - 2q))\, \mathbb{1}^{n \times n}_{\mathcal{R}^c \cap \mathcal{A}_0} + r(1 - 2q) \mathbb{1}^{n \times n}_{\mathcal{R}^c \cap \Gamma^{out}} \right] \quad (4.21)$$

$$f(\mathbf{E}, \mathbf{W}_0) = \lambda \left[ r(1 - 2q)\, \mathrm{sum}(\mathbf{E}_{\mathcal{R}^c}) \right] - \lambda \left[ \sum_{i=1}^{K} \left( r(2p_i - 1) - \frac{1}{\lambda n_i} \right)\, \mathrm{sum}(\mathbf{E}_{\mathcal{R}_{i,i}}) \right]$$

From $\mathbf{L}^0$ and the constraint $1 \geq \mathbf{L}_{i,j} \geq 0$, it follows that,

$$\mathbf{E}_{\mathcal{R}^c} \text{ is (entrywise) nonnegative.} \qquad (4.22)$$
$$\mathbf{E}_{\mathcal{R}} \text{ is (entrywise) nonpositive.}$$

Thus, $\mathrm{sum}(\mathbf{E}_{\mathcal{R}^c}) \leq 0$ and $\mathrm{sum}(\mathbf{E}_{\mathcal{R}_{i,i}}) \geq 0$. When $\lambda(2p_i - 1) - \frac{1}{n_i} \geq 0$ and $\lambda(2q - 1) \leq 0$; we will have $f(\mathbf{E}, \mathbf{W}_0) \geq 0$ for all feasible $\mathbf{E}$. This indeed holds due to the assumptions of Theorem 1 (see (4.1)), as we assumed $r(2p_i - 1) > \frac{1}{\lambda n_i}$ for $i = 1, 2 \cdots, K$ and $1 > 2q$.

We will now proceed to find a tight bound on the spectral norm of $\mathbf{W}^0$. We will say that random variable $X$ has a $\Delta(\zeta, \delta)$ distribution for $0 \leq \zeta, \delta \leq 1$, written as $X \sim \Delta(\zeta, \delta)$ if,

$$X = \begin{cases} 1 + \zeta(1 - 2\delta) & \text{w.p. } \zeta\delta \\ -1 + \zeta(1 - 2\delta) & \text{w.p. } \zeta(1 - \delta) \\ \zeta(1 - 2\delta) & \text{w.p. } 1 - \zeta \end{cases}$$

Variance of the above distribution is

$$\mathrm{Var}(X) = \zeta(1 - \zeta + 4\,\zeta\,\delta\,(1 - \delta)). \qquad (4.23)$$

**Theorem 3.** *Assume* $\mathbf{A} \in \mathbb{R}^{n \times n}$ *obeys the Stochastic Block Model* (2.1) *and let* $\mathbf{M} \in \mathbb{R}^{n \times n}$. *Let entries of* $\mathbf{M}$ *be as follows.*

$$\mathbf{M}_{i,j} \sim \begin{cases} \Delta(r, p_k) & \text{if } (i,j) \in \mathcal{R}_{k,k} \\ \Delta(r, q) & \text{if } (i,j) \in \mathcal{R}^c \end{cases}$$

*Then, for a constant* $\epsilon'$ *(to be determined) each of the following holds with probability* $1 - \exp(-\Omega(n))$.

- $\|\mathbf{M}\| \leq 2\sqrt{nr}\sqrt{1 - r + 4rq(1-q)}$
  $+ \max_{1 \leq i \leq K} 2\sqrt{n_i r}\sqrt{2(1-r) + 4r(q(1-q) + p_i(1-p_i))} + \epsilon'\sqrt{n}.$

- *Assume* $\max_{1 \leq i \leq K} n_i = o(n)$. *Then, for sufficiently large* $n$,

$$\|\mathbf{M}\| \leq (2\sqrt{r(1 - r + 4rq(1-q))} + \epsilon')\sqrt{n}.$$

*Proof.* For the first statement, let $\mathbf{M}_1$ be a random matrix with independent entries distributed as:
$$\mathbf{M}_1(i,j) \sim \Delta(r, q).$$
From standard results on random matrix theory [31], it follows that,
$$\|\mathbf{M}_1\| \leq (2\sqrt{r(1 - r + 4rq(1-q))} + \epsilon')\sqrt{n}$$
with the desired probability.

Also let $\mathbf{M}_2 = \mathbf{M} - \mathbf{M}_1$. We note that $\mathbf{M}_2$ is a block diagonal random matrix. Observe that $\mathbf{M}_2$ over $\mathcal{R}_{i,i}$, $\mathbf{M}_{2,\mathcal{R}_{i,i}}$ is sum of two independent random variables $\mathbf{M}_{\mathcal{R}_{i,i}} \sim \Delta(r, p_i)$ and $-\mathbf{M}_{1,\mathcal{R}_{i,i}} \sim \Delta(r, q)$. So, the variance is $2r(1-r) + 4r^2(q(1-q) + p_i(1-p_i))$. This similarly gives,
$$\|\mathbf{M}_{2,\mathcal{R}_{i,i}}\| \leq 2\sqrt{2r(1-r) + 4r^2(q(1-q) + p_i(1-p_i))}\sqrt{n_i} + \epsilon'\sqrt{n}$$

Now, observing, $\|\mathbf{M}_2\| = \sup_{1 \leq i \leq K} \|\mathbf{M}_{2,\mathcal{R}_{i,i}}\|$ and using a union bound over $i \leq K$ we have,
$$\|\mathbf{M}_2\| \leq \max_{1 \leq i \leq K} 2\sqrt{2r(1-r) + 4r^2(q(1-q) + p_i(1-p_i))}\sqrt{n_i} + \epsilon'\sqrt{n}$$

Finally, we use the triangle inequality $\|\mathbf{M}\| \leq \|\mathbf{M}_1\| + \|\mathbf{M}_2\|$ to conclude. ∎

The following lemma gives a bound on $\|\mathbf{W}_0\|$.

**Lemma 4.4.** *Recall that,* $\mathbf{W}_0$ *is a random matrix; where randomness is on the partially observed stochastic block model* $\mathbf{A_{obs}}$ *and it is given by,*

$$\mathbf{W}_0 = \lambda \left[ \sum_{i=1}^{K} (1 + r(1 - 2p_i)) \, \mathbb{1}^{n \times n}_{\mathcal{R}_{i,i} \cap \mathcal{A}_1} + (1 - r(1 - 2p_i)) \, \mathbb{1}^{n \times n}_{\mathcal{R}_{i,i} \cap \mathcal{A}_0} + r(1 - 2p_i)\mathbb{1}^{n \times n}_{\mathcal{R}_{i,i} \cap \Gamma^{out}} \right]$$
$$+ \lambda \left[ (1 + r(1 - 2q)) \, \mathbb{1}^{n \times n}_{\mathcal{R}^c \cap \mathcal{A}_1} + (-1 + r(1 - 2q)) \, \mathbb{1}^{n \times n}_{\mathcal{R}^c \cap \mathcal{A}_0} + r(1 - 2q)\mathbb{1}^{n \times n}_{\mathcal{R}^c \cap \Gamma^{out}} \right]$$

*Then, for any* $\epsilon' > 0$, *with probability* $1 - \exp(-\Omega(n))$, *we have*

$$\|\frac{1}{\lambda}\mathbf{W}_0\| \leq 2\sqrt{nr}\sqrt{1 - r + 4rq(1-q)} + \max_{1 \leq i \leq K} 2\sqrt{n_i r}\sqrt{2(1-r) + 4r(q(1-q) + p_i(1-p_i))} + \epsilon'\sqrt{n}$$

*Further, if* $\max_{1 \leq i \leq K} n_i = o(n)$. *Then, for sufficiently large* $n$, *with the same probability,*

$$\|\mathbf{W}_0\| \leq 2\lambda\sqrt{nr}\sqrt{1 - r + 4rq(1-q)} + \epsilon'\lambda\sqrt{n}.$$

*Proof.* $\frac{1}{\lambda}\mathbf{W}_0$ is a random matrix whose entries are i.i.d. and distributed as $\Delta(r, p_i)$ on $\mathcal{R}_{i,i}$ and $\Delta(r, q)$ on $\mathcal{R}^c$. Consequently, using Theorem 3 we obtain the result. ∎

Lemma 4.4 verifies that asymptotically with high probability we can make $\|\mathbf{W}_0\| < 1$ as long as $\lambda$ is sufficiently small. However, $\mathbf{W}_0$ itself is not sufficient for construction of the desired $\mathbf{W}$, since we do not have any guarantee that $\mathbf{W}_0 \in \mathcal{M}_\mathbf{U}$. In order to achieve this, we will *correct* $\mathbf{W}_0$ by projecting it onto $\mathcal{M}_\mathbf{U}$. Following lemma suggests that $\mathbf{W}_0$ does not change much by such a correction.

#### 4.3.2 Correcting the candidate $\mathbf{W}_0$

**Lemma 4.5.** $\mathbf{W}_0$ *is as described previously in* (4.21). *Let* $\mathbf{W}^H$ *be the projection of* $\mathbf{W}_0$ *on* $\mathcal{M}_{\mathbf{U}}$. *Then*

- $\|\mathbf{W}^H\| \leq \|\mathbf{W}_0\|$

- *For any* $\epsilon'' > 0$ *(constant to be determined), with probability* $1 - 6n^2 \exp(-2\epsilon''^2 n_{min})$ *we have*

$$\|\mathbf{W}_0 - \mathbf{W}^H\|_\infty \leq 3\lambda\epsilon''$$

*Proof.* Choose arbitrary vectors $\{\mathbf{u}_i\}_{i=K+1}^n$ to make $\{\mathbf{u}_i\}_{i=1}^n$ an orthonormal basis in $\mathbb{R}^n$. Call $\mathbf{U}_2 = [\mathbf{u}_{K+1} \ \ldots \ \mathbf{u}_n]$ and $\mathbf{P} = \mathbf{U}\mathbf{U}^T$, $\mathbf{P}_2 = \mathbf{U}_2\mathbf{U}_2^T$. Now notice that for any matrix $\mathbf{X} \in \mathbb{R}^{n \times n}$, $\mathbf{P}_2\mathbf{X}\mathbf{P}_2$ is in $\mathcal{M}_{\mathbf{U}}$ since $\mathbf{U}^T\mathbf{U}_2 = 0$. Let $\mathbf{I}$ denote the identity matrix. Then,

$$\begin{aligned}
\mathbf{X} - \mathbf{P}_2\mathbf{X}\mathbf{P}_2 &= \mathbf{X} - (\mathbf{I} - \mathbf{P})\mathbf{X}(\mathbf{I} - \mathbf{P}) \\
&= \mathbf{P}\mathbf{X} + \mathbf{X}\mathbf{P} - \mathbf{P}\mathbf{X}\mathbf{P} \in \mathcal{M}_{\mathbf{U}}^\perp
\end{aligned} \tag{4.24}$$

Hence, $\mathbf{P}_2\mathbf{X}\mathbf{P}_2$ is the orthogonal projection on $\mathcal{M}_{\mathbf{U}}$. Clearly,

$$\|\mathbf{W}^H\| = \|\mathbf{P}_2\mathbf{W}_0\mathbf{P}_2\| \leq \|\mathbf{P}_2\|^2\|\mathbf{W}_0\| \leq \|\mathbf{W}_0\|$$

For analysis of $\|\mathbf{W}_0 - \mathbf{W}^H\|_\infty$ we can consider terms on the right hand side of (4.24) separately as we have:

$$\|\mathbf{W}_0 - \mathbf{W}^H\|_\infty \leq \|\mathbf{P}\mathbf{W}_0\|_\infty + \|\mathbf{W}_0\mathbf{P}\|_\infty + \|\mathbf{P}\mathbf{W}_0\mathbf{P}\|_\infty$$

Clearly $\mathbf{P} = \sum_{i=1}^K \frac{1}{n_i} \mathbb{1}_{\mathbb{R}_{i,i}}^{n \times n}$. Then, each entry of $\frac{1}{\lambda}\mathbf{P}\mathbf{W}_0$ is either a summation of $n_i$ i.i.d. $\Delta(r, p_i)$ or $\Delta(r, q)$ mean zero random variables scaled by $n_i^{-1}$ for some $i \leq K$ or 0. Hence any $c, d \in [n]$ and $\epsilon'' > 0$

$$\mathbb{P}[|(\mathbf{P}\mathbf{W}_0)_{c,d}| \geq \lambda\epsilon''] \leq 2\exp(-2\epsilon''^2 n_{min})$$

Same (or better) bounds holds for entries of $\mathbf{W}_0\mathbf{P}$ and $\mathbf{P}\mathbf{W}_0\mathbf{P}$. Then a union bound over all entries of the three matrices will give with probability $1 - 6n^2 \exp(-2\epsilon''^2 n_{min})$, we have $\|\mathbf{W}_0 - \mathbf{W}^H\|_\infty \leq 3\lambda\epsilon''$. ∎

Recall that, $\gamma_{\text{succ}} := \max_{1 \leq i \leq K} 2r\sqrt{n_i}\sqrt{2(\frac{1}{r} - 1) + 4(q(1-q) + p_i(1-p_i))}$, and

$\mathbf{\Lambda}_{\text{succ}}^{-1} := 2r\sqrt{n}\sqrt{\frac{1}{r} - 1 + 4q(1-q)} + \gamma_{\text{succ}}$.

We can summarize our discussion so far in the following lemma,

**Lemma 4.6.** $\mathbf{W}_0$ *is as described previously in* (4.21). *Choose* $\mathbf{W}$ *to be projection of* $\mathbf{W}_0$ *on* $\mathcal{M}_{\mathbf{U}}$. *Also suppose* $\lambda \leq (1 - \delta)\mathbf{\Lambda}_{succ}$. *Then, with probability* $1 - 6n^2 \exp(-\Omega(n_{min})) - 4\exp(-\Omega(n))$ *we have,*

- $\|\mathbf{W}\| < 1$

- *For all feasible* $\mathbf{E}$, $f(\mathbf{E}, \mathbf{W}) \geq 0$.

*Proof.* To begin with, observe that $\mathbf{\Lambda}_{\text{succ}}^{-1}$ is $\Omega(\sqrt{n})$. Since $\lambda \leq \Lambda_{\text{succ}}$, $\lambda\sqrt{n} = \mathcal{O}(1)$. Consequently, using $\lambda\Lambda_{\text{succ}}^{-1} < 1$ and applying Lemma 4.4, and choosing a sufficiently small $\epsilon' > 0$, we conclude with,

$$\|\mathbf{W}\| \leq \|\mathbf{W}_0\| < 1$$

with probability $1 - \exp(-\Omega(n))$ where the constant in the exponent depends on the constant $\epsilon' > 0$.

Next, from Lemma 4.5 with probability $1 - 6n^2 \exp(-\frac{2}{9}\epsilon''^2 n_{min})$ we have $\|\mathbf{W}_0 - \mathbf{W}\|_\infty \leq \lambda\epsilon''$. Then based on (5.10) for all $\mathbf{E}$, we have that,

$$
\begin{aligned}
f(\mathbf{E}, \mathbf{W}) &= f(\mathbf{E}, \mathbf{W}_0) - \langle \mathbf{W}_0 - \mathbf{W}, \mathbf{E} \rangle \\
&\geq f(\mathbf{E}, \mathbf{W}_0) - \lambda\epsilon'' \left(\text{sum}(\mathbf{E}_\mathcal{R}) - \text{sum}(\mathbf{E}_{\mathcal{R}^c})\right) \\
&= \lambda\left[(r(1-2q) - \epsilon'')\text{sum}(\mathbf{E}_{\mathcal{R}^c})\right] \\
&\quad - \lambda\sum_{i=1}^{K}\left[\left(r(2p_i - 1) - \frac{1}{\lambda n_i} - \epsilon''\right)\text{sum}(\mathbf{E}_{\mathcal{R}_{i,i}})\right] \\
&\geq 0
\end{aligned}
$$

where we chose $\epsilon''$ to be a sufficiently small constant. In particular, we set $\epsilon'' < \mathbf{D}_\mathcal{A}$, i.e., set $\epsilon'' < r(1 - 2q)$ and $\epsilon'' < r(2p_i - 1) - \frac{1}{\lambda n_i}$ for all $1 \leq i \leq K$.

Hence, by using a union bound $\mathbf{W}$ satisfies both of the desired conditions. ∎

**Summary so far:** Combining the last lemma with Lemma 4.3, with high probability, either there exists a dual vector $\mathbf{W}^*$ which ensures $f(\mathbf{E}, \mathbf{W}^*) > 0$ or $\mathbf{E} \in \mathcal{M}_\mathbf{U}^\perp$. If former, we are done. Hence, we need to focus on the latter case and show that for all perturbations $\mathbf{E} \in \mathcal{M}_\mathbf{U}^\perp$, the objective will strictly increase at $(\mathbf{L}^0, \mathbf{S}^0)$ with high probability.

### 4.4 Solving for $\mathbf{E}^L \in \mathcal{M}_\mathbf{U}^\perp$ case

Recall that,

$$
g(\mathbf{E}) = \sum_{i=1}^{K}\frac{1}{n_i}\text{sum}(\mathbf{E}_{R_{i,i}}) + \lambda\left(\text{sum}(\mathbf{E}_{\mathcal{A}_0}) - \text{sum}(\mathbf{E}_{\mathcal{A}_1})\right)
$$

Let us define,

$$
g_1(\mathbf{X}) := \sum_{i=1}^{K}\frac{1}{n_i}\text{sum}(\mathbf{X}_{\mathcal{R}_{i,i}}),
$$

$$
g_2(\mathbf{X}) := \text{sum}(\mathbf{X}_{\mathcal{A}_0}) - \text{sum}(\mathbf{X}_{\mathcal{A}_1}),
$$

so that, $g(\mathbf{X}) = g_1(\mathbf{X}) + \lambda g_2(\mathbf{X})$. Also let $\mathbf{V} = [\mathbf{v}_1 \ \dots \ \mathbf{v}_K]$ where $\mathbf{v}_i = \sqrt{n_i}\mathbf{u}_i$. Thus, $\mathbf{V}$ is basically obtained by, normalizing columns of $\mathbf{U}$ to make its nonzero entries 1. Assume $\mathbf{E} \in \mathcal{M}_\mathbf{U}^\perp$. Then, by definition of $\mathcal{M}_\mathbf{U}^\perp$, we can write,

$$
\mathbf{E} = \mathbf{V}\mathbf{M}^T + \mathbf{N}\mathbf{V}^T.
$$

Let $\mathbf{m}_i, \mathbf{n}_i$ denote $i$'th columns of $\mathbf{M}, \mathbf{N}$ respectively. From $\mathbf{L}^0$ and (1.3) it follows that

$$
\mathbf{E}_{\mathcal{R}^c} \text{ is (entrywise) nonnegative}
$$
$$
\mathbf{E}_\mathcal{R} \text{ is (entrywise) nonpositive}
$$

Now, we list some simple observations regarding structure of $\mathbf{E}$. We can write

$$
\mathbf{E} = \sum_{i=1}^{K}(\mathbf{v}_i\mathbf{m}_i^T + \mathbf{n}_i\mathbf{v}_i^T) = \sum_{i=1}^{K+1}\sum_{j=1}^{K+1}\mathbf{E}_{\mathcal{R}_{i,j}} \tag{4.25}
$$

Notice that only two components : $\mathbf{v}_i\mathbf{m}_i^T$ and $\mathbf{n}_j\mathbf{v}_j^T$, contribute to the term $\mathbf{E}_{\mathcal{R}_{i,j}}$.

Let $\mathbf{E}^{i,j} \in \mathbb{R}^{n_i \times n_j}$ which is $\mathbf{E}$ induced by entries on $\mathcal{R}_{i,j}$. Basically, $\mathbf{E}^{i,j}$ is same as $\mathbf{E}_{\mathcal{R}_{i,j}}$ when we get rid of trivial zero rows and zero columns. Then

$$
\mathbf{E}^{i,j} = \mathbb{1}^{n_i}(\mathbf{m}_i^{\mathcal{C}_j})^T + \mathbf{n}_j^{\mathcal{C}_i}\mathbb{1}^{n_j\,T} \tag{4.26}
$$

where $\mathbf{m}_i^{\mathcal{C}_j}$ is the vector corresponding to the entries of $\mathcal{C}_j$ in $\mathbf{m}_i$. Similarly, $\mathbf{n}_j^{\mathcal{C}_i}$ is the vector corresponding to the entries of $\mathcal{C}_i$ in $\mathbf{n}_j$.

Clearly, given $\{\mathbf{E}^{i,j}\}_{1\leq i,j\leq n}$, $\mathbf{E}$ is uniquely determined. Now, assume we fix sum($\mathbf{E}^{i,j}$) for all $i,j$ and we would like to find the *worst* $\mathbf{E}$ subject to these constraints. Variables in such an optimization are $\mathbf{m}_i, \mathbf{n}_i$. Basically we are interested in,

$$\min g(\mathbf{E}) \tag{4.27}$$

$$\text{subject to}$$

$$\text{sum}(\mathbf{E}^{i,j}) = c_{i,j} \text{ for all } i,j$$

$$\mathbf{E}^{i,j} \begin{cases} \text{nonnegative if } i \neq j \\ \text{nonpositive if } i = j \end{cases} \tag{4.28}$$

where $\{c_{i,j}\}$ are constants. Constraint (4.28) follows from (5.10).

**Remark:** For the special case of $i = j = K + 1$, notice that $\mathbf{E}^{i,j} = 0$.

In (4.27), $g_1(\mathbf{E})$ is fixed and is equal to $\sum_{i=1}^{K} \frac{1}{n_i} c_{i,i}$. Consequently, we just need to do the optimization with the objective $g_2(\mathbf{E}) = \text{sum}(\mathbf{E}_{\mathcal{A}_0}) - \text{sum}(\mathbf{E}_{\mathcal{A}_1})$.

Let $\beta_{i,j} \subseteq [n_i] \times [n_j]$ be a set of coordinates defined as follows. For any $(c,d) \in [n_i] \times [n_j]$

$$(c,d) \in \beta_{i,j} \text{ iff } (a_{i,c}, a_{j,d}) \in \mathcal{A}$$

For $(i_1, j_1) \neq (i_2, j_2)$, $(\mathbf{m}_{i_1}^{\mathcal{C}_{j_1}}, \mathbf{n}_{j_1}^{\mathcal{C}_{i_1}})$ and $(\mathbf{m}_{i_2}^{\mathcal{C}_{j_2}}, \mathbf{n}_{j_2}^{\mathcal{C}_{i_2}})$ are independent variables. Consequently, due to (4.26), we can partition problem (4.27) into the following smaller disjoint problems.

$$\min_{\mathbf{m}_i^j, \mathbf{n}_j^i} \text{sum}(\mathbf{E}_{\beta_{i,j}^c}^{i,j}) - \text{sum}(\mathbf{E}_{\beta_{i,j}}^{i,j}) \tag{4.29}$$

$$\text{subject to}$$

$$\text{sum}(\mathbf{E}^{i,j}) = c_{i,j}$$

$$\mathbf{E}^{i,j} \text{ is } \begin{cases} \text{nonnegative if } i \neq j \\ \text{nonpositive if } i = j \end{cases}$$

Then, we can solve these problems locally (for each $i,j$) to finally obtain,

$$g_2(\mathbf{E}^{L,*}) = \sum_{i,j} \text{sum}(\mathbf{E}_{\beta_{i,j}^c}^{i,j,*}) - \sum_{i,j} \text{sum}(\mathbf{E}_{\beta_{i,j}}^{i,j,*})$$

to find the overall result of problem (4.27), where $*$ denotes the optimal solutions in problems (4.27) and (4.29). The following lemma will be useful for analysis of these local optimizations.

**Lemma 4.7.** *Let $\mathbf{a} \in \mathbb{R}^c$, $\mathbf{b} \in \mathbb{R}^d$ and $X = \mathbb{1}^c \mathbf{b}^T + \mathbf{a} \mathbb{1}^{d^T}$ be variables and $C_0 \geq 0$ be a constant. Also let $\beta \subseteq [c] \times [d]$. Consider the following optimization problem*

$$\min_{\mathbf{a},\mathbf{b}} \text{ sum}(\mathbf{X}_\beta)$$

$$\text{subject to}$$

$$\mathbf{X}_{i,j} \geq 0 \text{ for all } i,j$$

$$\text{sum}(\mathbf{X}) = C_0$$

*For this problem there exists a (entrywise) nonnegative minimizer $(\mathbf{a}^0, \mathbf{b}^0)$.*

*Proof.* Let $x_i$ denotes $i$'th entry of vector $\mathbf{x}$. Assume $(\mathbf{a}^*, \mathbf{b}^*)$ is a minimizer. Without loss of generality assume $b_1^* = \min_{i,j}\{a_i^*, b_j^*\}$. If $b_1^* \geq 0$ we are done. Otherwise, since $\mathbf{X}_{i,j} \geq 0$ we have $a_i^* \geq -b_1^*$ for all $i \leq c$. Then set $\mathbf{a}^0 = \mathbf{a}^* + \mathbb{1}^c b_1^*$ and $\mathbf{b}^0 = \mathbf{b}^* - \mathbb{1}^d b_1^*$. Clearly, $(\mathbf{a}^0, \mathbf{b}^0)$ is nonnegative. On the other hand, we have:

$$\mathbf{X}^* = \mathbb{1}^c \mathbf{b}^{*T} + \mathbf{a}^* \mathbb{1}^{d^T} = \mathbb{1}^c \mathbf{b}^{0T} + \mathbf{a}^0 \mathbb{1}^{d^T} = \mathbf{X}^0,$$

which implies,

$$\text{sum}(\mathbf{X}_\beta^*) = \text{sum}(\mathbf{X}_\beta^0) = \text{optimal value}$$

∎

**Lemma 4.8.** *A direct consequence of Lemma 4.7 is the fact that in the local optimizations* (4.29), *Without loss of generality, we can assume* $(\mathbf{m}_i^{\mathcal{C}_j}, \mathbf{n}_j^{\mathcal{C}_i})$ *entrywise nonnegative whenever* $i \neq j$ *and entrywise nonpositive when* $i = j$. *This follows from the structure of* $\mathbf{E}^{i,j}$ *given in* (4.26) *and* (5.10).

The following lemma will help us characterize the relationship between $\mathrm{sum}(\mathbf{E}^{i,j})$ and $\mathrm{sum}(\mathbf{E}^{i,j}_{\beta_{i,j}^c})$.

**Lemma 4.9.** *Let* $\beta$ *be a random set generated by choosing elements of* $[c] \times [d]$ *independently with probability* $0 \leq \eta \leq 1$. *Then for any* $\epsilon' > 0$ *with probability* $1 - d\exp(-2\epsilon'^2 c)$ *for all nonzero and entrywise nonnegative* $\mathbf{a} \in \mathbb{R}^d$ *we'll have:*

$$\mathrm{sum}(\mathbf{X}_\beta) > (\eta - \epsilon')\mathrm{sum}(\mathbf{X}) \tag{4.30}$$

*where* $\mathbf{X} = \mathbb{1}^c \mathbf{a}^T$. *Similarly, with the same probability, for all such* $\mathbf{a}$, *we'll have* $\mathrm{sum}(\mathbf{X}_\beta) < (\eta + \epsilon')\mathrm{sum}(\mathbf{X})$

*Proof.* We'll only prove the first statement (4.30) as the proofs are identical. For each $i \leq d$, $a_i$ occurs exactly $c$ times in $\mathbf{X}$ as $i$'th column of $X$ is $\mathbb{1}^c a_i$. By using a Chernoff bound, we can estimate the number of coordinates of $i$'th column which are element of $\beta$ (call this number $C_i$) as we can view this number as a sum of $c$ i.i.d. Bernoulli$(\eta)$ random variables. Then

$$\mathbb{P}(C_i \leq c(\eta - \epsilon')) \leq \exp(-2\epsilon'^2 c)$$

Now, we can use a union bound over all columns to make sure for all $i$, $C_i > c(\eta - \epsilon')$

$$\mathbb{P}(C_i > c(r - \epsilon') \text{ for all } i \leq d) \geq 1 - d\exp(-2\epsilon'^2 c)$$

On the other hand if each $C_i > c(\eta - \epsilon')$ then for any nonnegative $\mathbf{a} \neq 0$,

$$\mathrm{sum}(\mathbf{X}_\beta) = \sum_{(i,j)\in\beta} \mathbf{X}_{i,j} = \sum_{i=}^{d} C_i a_i > c(\eta - \epsilon') \sum_{i=1}^{d} a_i = (\eta - \epsilon')\mathrm{sum}(\mathbf{X})$$

∎

Using Lemma 4.9, we can calculate a lower bound for $g(\mathbf{E})$ with high probability as long as the cluster sizes are sufficiently large. Due to (4.25) and the linearity of $g(\mathbf{E})$, we can focus on contributions due to specific clusters i.e. $\mathbf{v}_i \mathbf{m}_i^T + \mathbf{n}_i \mathbf{v}_i^T$ for the $i$'th cluster. We additionally know the simple structure of $\mathbf{m}_i, \mathbf{n}_i$ from Lemma 4.8. In particular, subvectors $\mathbf{m}_i^{\mathcal{C}_i}$ and $\mathbf{n}_i^{\mathcal{C}_i}$ of $\mathbf{m}_i, \mathbf{n}_i$ can be assumed to be nonpositive and rest of the entries are nonnegative.

**Lemma 4.10.** *Assume,* $1 \leq l \leq K$, $\mathbf{D}_\mathcal{A} > 0$. *Then, with probability* $1 - n\exp(-2\mathbf{D}_\mathcal{A}^2(n_l - 1))$, *we have* $g(\mathbf{v}_l \mathbf{m}_l^T) \geq 0$ *for all* $\mathbf{m}_l$. *Also, if* $\mathbf{m}_l \neq 0$ *then inequality is strict.*

*Proof.* Recall that $\mathbf{m}_l$ satisfies $\mathbf{m}_l^{\mathcal{C}_i}$ is nonpositive/nonnegative when $i = l / i \neq l$ for all $i$. Define $\mathbf{X}^i := \mathbb{1}^{n_l} \mathbf{m}_l^{\mathcal{C}_i T}$. We can write

$$g(\mathbf{v}_l \mathbf{m}_l^T) = \frac{1}{n_l}\mathrm{sum}(\mathbf{X}^l) + \sum_{i=1}^{K} \lambda h(\mathbf{X}^i, \beta_{l,i}^c)$$

where $h(\mathbf{X}^i, \beta_{l,i}^c) = \mathrm{sum}(\mathbf{X}^i_{\beta_{l,i}^c}) - \mathrm{sum}(\mathbf{X}^i_{\beta_{l,i}})$. Now assume $i \neq l$. Using Lemma 4.9 and the fact that $\beta_{l,i}$ is a randomly generated subset (with parameter $q$), with probability $1 - n_i\exp(-2\epsilon'^2 n_l)$, for all $\mathbf{X}^i$, we have,

$$h(\mathbf{X}^i, \beta_{l,i}^c) \geq (r(1-q) - \epsilon')\mathrm{sum}(\mathbf{X}^i) - (rq + \epsilon')\mathrm{sum}(\mathbf{X}^i)$$
$$= (r(1 - 2q) - 2\epsilon')\mathrm{sum}(\mathbf{X}^i)$$

where inequality is strict if $X^i \neq 0$. Similarly, when $i = l$ with probability at least $1 - n_l\exp(-2\epsilon'^2(n_l - 1))$, we have,

$$\frac{1}{\lambda n_l}\mathrm{sum}(\mathbf{X}^l) + h(\mathbf{X}^l, \beta_{l,l}^c) \geq \left(r(1 - p_l) + \epsilon' + \frac{1}{\lambda n_l}\right)\mathrm{sum}(\mathbf{X}^l) - (rp_l - \epsilon')\mathrm{sum}(\mathbf{X}^l)$$
$$= -\left(r(2p_l - 1) - \frac{1}{\lambda n_l} - 2\epsilon'\right)\mathrm{sum}(\mathbf{X}^l)$$

Choosing $\epsilon' = \frac{\mathbf{D}_{\mathcal{A}}}{2}$ and using the facts that $r(1 - 2q) - 2\mathbf{D}_{\mathcal{A}} \geq 0$, $r(2p_l - 1) - \frac{1}{\lambda n_l} - 2\mathbf{D}_{\mathcal{A}} \geq 0$ and using a union bound, with probability $1 - n\exp(-2\mathbf{D}_{\mathcal{A}}^2(n_l - 1))$, we have $g(\mathbf{v}_l\mathbf{m}_l^T) \geq 0$ and the inequality is strict when $\mathbf{m}_l \neq 0$ as at least one of the $\mathbf{X}^i$'s will be nonzero. $\blacksquare$

The following lemma immediately follows from Lemma 4.10 and summarizes the main result of the section.

**Lemma 4.11.** *Let* $\mathbf{D}_{\mathcal{A}}$ *be as defined in* (4.1) *and assume* $\mathbf{D}_{\mathcal{A}} > 0$. *Then with probability* $1 - 2nK\exp(-2\mathbf{D}_{\mathcal{A}}^2(n_{min} - 1))$ *we have* $g(\mathbf{E}^L) > 0$ *for all nonzero feasible* $\mathbf{E}^L \in \mathcal{M}_U^{\perp}$.

### 4.5 The Final Step

**Lemma 4.12.** *Let* $p_{min} > \frac{1}{2} > q$ *and* $\mathcal{G}$ *be a random graph generated according to Model* 2.1 *and* 2.2 *with cluster sizes* $\{n_i\}_{i=1}^K$. *If* $\lambda \leq (1 - \epsilon)\mathbf{\Lambda}_{succ}$ *and* $\mathbf{D}_{\min} = \min\limits_{1 \leq i \leq n} r(2p_i - 1)n_i \geq (1 + \epsilon)\frac{1}{\lambda}$, *then* $(\mathbf{L}^0, \mathbf{S}^0)$ *is the unique optimal solution to Program* 1.1 *with probability* $1 - \exp(-\Omega(n)) - 6n^2\exp(-\Omega(n_{min}))$.

*Proof.* Based on Lemma 4.6 and Lemma 4.11,
with probability $1 - cn^2\exp(-C\left(\min\{r(1 - 2q), r(2p_{min} - 1)\}\right)^2 n_{min})$,

- There exists $\mathbf{W} \in \mathcal{M}_{\mathbf{U}}$ with $\|\mathbf{W}\| < 1$ such that for all feasible $\mathbf{E}^L$, $f(\mathbf{E}^L, \mathbf{W}) \geq 0$.
- For all nonzero $\mathbf{E}^L \in \mathcal{M}_{\mathbf{U}}^{\perp}$ we have $g(\mathbf{E}^L) > 0$.

Consequently based on Lemma 4.3, $(\mathbf{L}^0, \mathbf{S}^0)$ is the unique optimal of Problem 1.1. $\blacksquare$

## 5 Proof of Results for Improved Convex Program

This section will show that, the optimal solution of Problem 1.4 is the pair $(\mathbf{L}^0, \mathbf{S}^0)$ under reasonable conditions, where,

$$\mathbf{L}^0 = \mathbb{1}_{\mathcal{R}}^{n \times n}, \ \mathbf{S}^0 = \mathbf{S}_{\mathbf{obs}}^0 = \mathbb{1}_{\mathcal{R} \cap \mathcal{A}_0}^{n \times n} \tag{5.1}$$

Also denote the true optimal pair by $(\mathbf{L}^*, \mathbf{S}^*)$. Let $1 \geq p_{min} > q > 0$ and $0 \leq r \leq 1$. $\mathcal{G}$ be a random graph generated according to the stochastic block model 2.1 with cluster sizes $\{n_i\}_{i=1}^K$. Let the observation model be as defined in (2.2). Theorem 2 is based on the following lemma:

**Lemma 5.1.** *If* $\lambda < \tilde{\mathbf{\Lambda}}_{succ}$ *and* $\tilde{\mathbf{D}}_{\min} > \frac{1}{\lambda}$, *then* $(\mathbf{L}^0, \mathbf{S}^0)$ *is the unique optimal solution to Program* 1.4 *with high probability.*

Given $q, \{p_i\}_{i=1}^K$, define the following parameter which will be useful for the subsequent analysis. This parameter can be seen as a measure of distinctness of the "worst" cluster from the "background noise". Here, by background noise we mean the edges over $\mathcal{R}^c$.

$$\tilde{\mathbf{D}}_{\mathcal{A}} = \frac{1}{2}\min\left\{r(1 - q), \left\{r(p_i - q) - \frac{1}{\lambda n_i}\right\}_{i=1}^K\right\} \tag{5.2}$$

$$= \frac{1}{2}\min\left\{r(1 - q), \frac{\tilde{\mathbf{D}}_i - \lambda^{-1}}{n_i}\right\}$$

### 5.1 Perturbation Analysis

Our aim is to show that $(\mathbf{L}^0, \mathbf{S}^0)$ defined in (5.1) is unique optimal solution to Problem 1.4.

**Lemma 5.2.** *Let* $(\mathbf{E}^L, \mathbf{E}^S)$ *be a feasible perturbation. Then, the objective will increase by at least,*

$$f(\mathbf{E}^L, \mathbf{W}) = \sum_{i=1}^K \frac{1}{n_i} sum(\mathbf{E}_{\mathcal{R}_{i,i}}^L) + \langle \mathbf{E}^L, \mathbf{W}\rangle + \lambda sum(\mathbf{E}_{\mathcal{A}_0}^L) \tag{5.3}$$

*for any* $\mathbf{W} \in \mathcal{M}, \|\mathbf{W}\| \leq 1$.

*Proof.* From constraint (1.6), we have $\mathbf{L}_{i,j} = \mathbf{S}_{i,j}$ whenever $\mathbf{A}_{i,j}^{obs} = 0$. So, $\mathbf{L}_{\mathcal{A}_0}^* = \mathbf{S}_{\mathcal{A}_0}^*$. Further, recall that $\mathbf{S}$ can be split as $\mathbf{S} = \mathbf{S_{obs}} + \mathbf{S_{rest}}$, where $\mathbf{S_{rest}}$ denotes the entries of $\mathbf{S}$ other than those corresponding to the observed entries of $\mathbf{A}$. Furthermore, we claim that at the optimal, $\mathbf{S_{rest}} = 0$, since if otherwise, the objective can be strictly decreased by setting $\mathbf{S_{rest}} = 0$. So, without loss of generality,

$$\mathbf{S}^* = \mathbf{L}_{\mathcal{A}_0}^*. \tag{5.4}$$

Recall that,

$$\|\mathbf{L}^0 + \mathbf{E}^L\|_\star + \lambda \|\mathbf{S}^0 + \mathbf{E}^S\|_1 - (\|\mathbf{L}^0\|_\star + \lambda \|\mathbf{S}^0\|_1) \geq \langle \partial\|\mathbf{L}^0\|_\star, \mathbf{E}^L\rangle + \lambda\langle\partial\|\mathbf{S}^0\|_1, \mathbf{E}^S\rangle$$
$$= \langle \mathbf{U}\mathbf{U}^T + \mathbf{W}, \mathbf{E}^L\rangle + \lambda\langle\mathrm{sign}(\mathbf{S}^0) + \mathbf{Q}, \mathbf{E}^S\rangle$$

Using $\mathrm{sign}(\mathbf{S}^0) = \mathbb{1}_{\mathcal{A}_0\cap\mathcal{R}}^{n\times n}$, and choosing $\mathbf{Q} = \mathbb{1}_{\mathcal{A}_0-(\mathcal{A}_0\cap\mathcal{R})}^{n\times n}$, we get,

$$\|\mathbf{L}^0 + \mathbf{E}^L\|_\star + \lambda \|\mathbf{S}^0 + \mathbf{E}^S\|_1 - (\|\mathbf{L}^0\|_\star + \lambda \|\mathbf{S}^0\|_1) \geq \langle \mathbf{W}, \mathbf{E}^L\rangle$$
$$+ \underbrace{\sum_{i=1}^{K} \frac{1}{n_i}\mathrm{sum}(\mathbf{E}_{R_{i,i}}^L) + \lambda\left(\mathrm{sum}(\mathbf{E}_{\mathcal{A}_0}^L)\right)}_{:=g(\mathbf{E}^L)}$$

$$\tag{5.5}$$

for any $\mathbf{W} \in \mathcal{M}$. ∎

From this point onward, for simplicity we will ignore the superscript L on $\mathbf{E}^L$ and just use $\mathbf{E}$.

Define,

$$g(\mathbf{E}) := \sum_{i=1}^{K} \frac{1}{n_i}\mathrm{sum}(\mathbf{E}_{\mathcal{R}_{i,i}}) + \lambda\mathrm{sum}(\mathbf{E}_{\mathcal{A}_0})). \tag{5.6}$$

Also, define $f(\mathbf{E}, \mathbf{W}) := g(\mathbf{E}) + \langle\mathbf{W}, \mathbf{E}\rangle$. Our aim is to show that for all feasible perturbations $\mathbf{E}$, there exists $\mathbf{W}$ such that,

$$f(\mathbf{E}, \mathbf{W}) = g(\mathbf{E}) + \langle\mathbf{W}, \mathbf{E}\rangle > 0. \tag{5.7}$$

Note that $g(\mathbf{E})$ does not depend on $\mathbf{W}$.

We can directly use Lemma 4.3. So, as in the previous section, we have broken down our aim into two steps.

1. Construct $\mathbf{W} \in \mathcal{M}_\mathbf{U}$ with $\|\mathbf{W}\| < 1$, such that $f(\mathbf{E}, \mathbf{W}) \geq 0$ for all feasible perturbations $\mathbf{E}$.

2. For all non-zero feasible $\mathbf{E} \in \mathcal{M}_\mathbf{U}^\perp$, show that $g(\mathbf{E}) > 0$.

As a first step, in Section 5.2, we will argue that, under certain conditions, there exists a $\mathbf{W} \in \mathcal{M}_\mathbf{U}$ with $\|\mathbf{W}\| < 1$ such that with high probability, $f(\mathbf{E}, \mathbf{W}) \geq 0$ for all feasible $\mathbf{E}$. Recall that such a $\mathbf{W}$ is called the dual certificate. Secondly, in Section 5.3, we will show that, under certain conditions, for all $\mathbf{E} \in \mathcal{M}_\mathbf{U}^\perp$ with high probability, $g(\mathbf{E}) > 0$. Finally, combining these two arguments, and using Lemma 4.3 we will conclude that $(\mathbf{L}^0, \mathbf{S}^0)$ is the unique optimal with high probability.

## 5.2 Showing existence of the dual certificate

Recall that

$$f(\mathbf{E}, \mathbf{W}) = \sum_{i=1}^{K} \frac{1}{n_i}\mathrm{sum}(\mathbf{E}_{\mathcal{R}_{i,i}}) + \langle\mathbf{E}, \mathbf{W}\rangle + \lambda\mathrm{sum}(\mathbf{E}_{\mathcal{A}_0})$$

$\mathbf{W}$ will be constructed from the candidate $\mathbf{W}_0$, which is given as follows.

### 5.2.1 Candidate $\mathbf{W}_0$

Based on Program 1.4, we propose the following,

$$\mathbf{W}_0 = \sum_{i=1}^{K} c_i \mathbb{1}_{\mathcal{R}_{i,i}}^{n \times n} + c \mathbb{1}^{n \times n} - \lambda \mathbb{1}_{\mathcal{A}_0}^{n \times n}, \tag{5.8}$$

where $\{c_i\}_{i=1}^{K}, c$ are real numbers to be determined.

$$f(\mathbf{E}, \mathbf{W}^0) = \sum_{i=1}^{K} (\frac{1}{n_i} + c_i) \operatorname{sum}(\mathbf{E}_{\mathcal{R}_{i,i}}) + c \operatorname{sum}(\mathbf{E})$$

Note that $\mathbf{W}_0$ is a random matrix where randomness is due to $\mathbf{A}_{\mathbf{obs}}$. In order to ensure a small spectral norm, we will set its expectation to 0, i.e., we will choose $c, \{c_i\}'s$ to ensure that $\mathbb{E}[\mathbf{W}_0] = 0$.

Following from the partially observed Stochastic Block Model (Definition 2.1 and Definition 2.2), the expectation of an entry of $\mathbf{W}_0$ on $\mathcal{R}_{i,i}$ (region corresponding to cluster $i$) and $\mathcal{R}^c$ (region outside the clusters) is $c_i + \lambda r(p_i - q)$ and $c + \lambda r(q-1)$ respectively. Hence, we set,

$$c_i = -\lambda r(p_i - q) \quad \text{and} \quad c = \lambda r(1 - q),$$

With these, choices, the candidate $\mathbf{W}_0$ and $f(\mathbf{E}, \mathbf{W}_0)$ take the following forms,

$$
\begin{aligned}
\mathbf{W}_0 &= \lambda \left[ \sum_{i=1}^{K} -(1 - r(1 - p_i)) \, \mathbb{1}_{\mathcal{R}_{i,i} \cap \mathcal{A}_0}^{n \times n} + r(1 - p) \left( \mathbb{1}_{\mathcal{R}_{i,i} \cap \mathcal{A}_1}^{n \times n} + \mathbb{1}_{\mathcal{R}_{i,i} \cap \Gamma^{out}}^{n \times n} \right) \right] \\
&\quad + \lambda \left[ -(1 - r(1 - q)) \mathbb{1}_{\mathcal{R}^c \cap \mathcal{A}_0}^{n \times n} + r(1 - q) \left( \mathbb{1}_{\mathcal{R}^c \cap \mathcal{A}_1}^{n \times n} + \mathbb{1}_{\mathcal{R}^c \cap \Gamma^{out}}^{n \times n} \right) \right]
\end{aligned} \tag{5.9}
$$

$$f(\mathbf{E}, \mathbf{W}_0) = \lambda \left[ r(1-q) \operatorname{sum}(\mathbf{E}) \right] - \lambda \left[ \sum_{i=1}^{K} \left( r(p_i - q) - \frac{1}{\lambda n_i} \right) \operatorname{sum}(\mathbf{E}_{\mathcal{R}_{i,i}}) \right]$$

From $\mathbf{L}^0$ and the constraint $1 \geq \mathbf{L}_{i,j} \geq 0$, it follows that,

$$\mathbf{E}_{\mathcal{R}^c} \text{ is (entrywise) nonnegative.} \tag{5.10}$$
$$\mathbf{E}_{\mathcal{R}} \text{ is (entrywise) nonpositive.}$$

Thus, $\operatorname{sum}(\mathbf{E}_{\mathcal{R}^c}) \leq 0$ and $\operatorname{sum}(\mathbf{E}_{\mathcal{R}_{i,i}}) \geq 0$. When $\lambda r(p_i - q) - \frac{1}{n_i} \geq 0$ and $\lambda(1-q) \geq 0$; we will have $f(\mathbf{E}, \mathbf{W}_0) \geq 0$ for all feasible $\mathbf{E}$. This indeed holds due to the assumptions of Theorem 2 (see (5.2)), as we assumed $r(p_i - q) > \frac{1}{\lambda n_i}$ for $i = 1, 2 \cdots, K$ and $1 > q$.

Using the same technique as in Theorem 3, we can bound the spectral norm of $\mathbf{W}^0$ as follows

**Lemma 5.3.** *Recall that, $\mathbf{W}_0$ is a random matrix; where randomness is on the partially observed stochastic block model $\mathbf{A}_{\mathbf{obs}}$ and it is given by,*

$$
\begin{aligned}
\mathbf{W}_0 &= \lambda \left[ \sum_{i=1}^{K} -(1 - r(1 - p_i)) \, \mathbb{1}_{\mathcal{R}_{i,i} \cap \mathcal{A}_0}^{n \times n} + r(1 - p) \left( \mathbb{1}_{\mathcal{R}_{i,i} \cap \mathcal{A}_1}^{n \times n} + \mathbb{1}_{\mathcal{R}_{i,i} \cap \Gamma^{out}}^{n \times n} \right) \right] \\
&\quad + \lambda \left[ -(1 - r(1 - q)) \mathbb{1}_{\mathcal{R}^c \cap \mathcal{A}_0}^{n \times n} + r(1 - q) \left( \mathbb{1}_{\mathcal{R}^c \cap \mathcal{A}_1}^{n \times n} + \mathbb{1}_{\mathcal{R}^c \cap \Gamma^{out}}^{n \times n} \right) \right]
\end{aligned}
$$

*Then, for any $\epsilon' > 0$, with probability $1 - \exp(-\Omega(n))$, we have*

$$\|\frac{1}{\lambda} \mathbf{W}_0\| \leq 2\sqrt{nr}\sqrt{(1-q)(1-r+rq)} + \max_{1 \leq i \leq K} 2\sqrt{n_i r}\sqrt{(1-p_i)(1-r+rp_i) + (1-q)(1-r+rq)} + \epsilon'\sqrt{n}$$

*Further, if $\max_{1 \leq i \leq K} n_i = o(n)$. Then, for sufficiently large $n$, with the same probability,*

$$\|\mathbf{W}_0\| \leq 2\lambda\sqrt{nr}\sqrt{(1-q)(1-r+rq)} + \epsilon'\lambda\sqrt{n}.$$

Lemma 5.3 verifies that asymptotically with high probability we can make $\|\mathbf{W}_0\| < 1$ as long as $\lambda$ is sufficiently small. However, $\mathbf{W}_0$ itself is not sufficient for construction of the desired $\mathbf{W}$, since we do not have any guarantee that $\mathbf{W}_0 \in \mathcal{M}_{\mathbf{U}}$. In order to achieve this, we will *correct* $\mathbf{W}_0$ by projecting it onto $\mathcal{M}_{\mathbf{U}}$. Lemma 4.5 can be used to here.

Recall that, $\tilde{\gamma}_{\mathrm{succ}} := 2 \max\limits_{1 \leq i \leq K} r\sqrt{n_i} \sqrt{(1-p_i)(\frac{1}{r}-1+p_i)+(1-q)(\frac{1}{r}-1+q)}$ and $\tilde{\mathbf{\Lambda}}_{\mathrm{succ}}^{-1} := 2r\sqrt{n}\sqrt{(\frac{1}{r}-1+q)(1-q)} + \tilde{\gamma}_{\mathrm{succ}}$.

We can summarize our discussion so far in the following lemma,

**Lemma 5.4.** $\mathbf{W}_0$ *is as described previously in* (5.9). *Choose* $\mathbf{W}$ *to be projection of* $\mathbf{W}_0$ *on* $\mathcal{M}_{\mathbf{U}}$. *Also suppose* $\lambda \leq (1-\delta)\tilde{\mathbf{\Lambda}}_{succ}$. *Then, with probability* $1 - 6n^2 \exp(-\Omega(n_{min})) - 4\exp(-\Omega(n))$ *we have,*

- $\|\mathbf{W}\| < 1$

- *For all feasible* $\mathbf{E}$, $f(\mathbf{E}, \mathbf{W}) \geq 0$.

*Proof.* To begin with, observe that $\tilde{\mathbf{\Lambda}}_{\mathrm{succ}}^{-1}$ is $\Omega(\sqrt{n})$. Since $\lambda \leq \tilde{\mathbf{\Lambda}}_{\mathrm{succ}}$, $\lambda\sqrt{n} = \mathcal{O}(1)$. Consequently, using $\lambda\tilde{\mathbf{\Lambda}}_{\mathrm{succ}}^{-1} < 1$ and applying Lemma 5.3, and choosing a sufficiently small $\epsilon' > 0$, we conclude with,

$$\|\mathbf{W}\| \leq \|\mathbf{W}_0\| < 1$$

with probability $1 - \exp(-\Omega(n))$ where the constant in the exponent depends on the constant $\epsilon' > 0$.

Next, from Lemma 4.5 with probability $1 - 6n^2 \exp(-\frac{2}{9}\epsilon''^2 n_{min})$ we have $\|\mathbf{W}_0 - \mathbf{W}\|_\infty \leq \lambda\epsilon''$. Then based on (5.10) for all $\mathbf{E}$, we have that,

$$
\begin{aligned}
f(\mathbf{E}, \mathbf{W}) &= f(\mathbf{E}, \mathbf{W}_0) - \langle \mathbf{W}_0 - \mathbf{W}, \mathbf{E} \rangle \\
&\geq f(\mathbf{E}, \mathbf{W}_0) - \lambda\epsilon'' \left( \mathrm{sum}(\mathbf{E}_\mathcal{R}) - \mathrm{sum}(\mathbf{E}_{\mathcal{R}^c}) \right) \\
&= \lambda \left[ (r(1-q) - \epsilon'')\mathrm{sum}(\mathbf{E}_{\mathcal{R}^c}) \right] \\
&\quad - \lambda \sum_{i=1}^{K} \left[ \left( r(p_i - q) - \frac{1}{\lambda n_i} - \epsilon'' \right) \mathrm{sum}(\mathbf{E}_{\mathcal{R}_{i,i}}) \right] \\
&\geq 0
\end{aligned}
$$

where we chose $\epsilon''$ to be a sufficiently small constant. In particular, we set $\epsilon'' < \tilde{\mathbf{D}}_\mathcal{A}$, i.e., set $\epsilon'' < r(1-q)$ and $\epsilon'' < r(p_i - q) - \frac{1}{\lambda n_i}$ for all $1 \leq i \leq K$.

Hence, by using a union bound $\mathbf{W}$ satisfies both of the desired conditions. ∎

**Summary so far:** Combining the last lemma with Lemma 4.3, with high probability, either there exists a dual vector $\mathbf{W}^*$ which ensures $f(\mathbf{E}, \mathbf{W}^*) > 0$ or $\mathbf{E} \in \mathcal{M}_{\mathbf{U}}^\perp$. If former, we are done. Hence, we need to focus on the latter case and show that for all perturbations $\mathbf{E} \in \mathcal{M}_{\mathbf{U}}^\perp$, the objective will strictly increase at $(\mathbf{L}^0, \mathbf{S}^0)$ with high probability.

### 5.3 Solving for $\mathbf{E}^L \in \mathcal{M}_{\mathbf{U}}^\perp$ case

Recall that,

$$g(\mathbf{E}) = \sum_{i=1}^{K} \frac{1}{n_i} \mathrm{sum}(\mathbf{E}_{R_{i,i}}) + \lambda\mathrm{sum}(\mathbf{E}_{\mathcal{A}_0})$$

Let us define,

$$g_1(\mathbf{X}) := \sum_{i=1}^{K} \frac{1}{n_i} \mathrm{sum}(\mathbf{X}_{\mathcal{R}_{i,i}}),$$

$$g_2(\mathbf{X}) := \mathrm{sum}(\mathbf{X}_{\mathcal{A}_0}),$$

so that, $g(\mathbf{X}) = g_1(\mathbf{X}) + \lambda g_2(\mathbf{X})$. Also let $\mathbf{V} = [\mathbf{v}_1 \ \ldots \ \mathbf{v}_K]$ where $\mathbf{v}_i = \sqrt{n_i}\mathbf{u}_i$. Thus, $\mathbf{V}$ is basically obtained by, normalizing columns of $\mathbf{U}$ to make its nonzero entries 1. Assume $\mathbf{E} \in \mathcal{M}_{\mathbf{U}}^{\perp}$. Then, by definition of $\mathcal{M}_{\mathbf{U}}^{\perp}$, we can write,

$$\mathbf{E} = \mathbf{V}\mathbf{M}^T + \mathbf{N}\mathbf{V}^T.$$

Let $\mathbf{m}_i, \mathbf{n}_i$ denote $i$'th columns of $\mathbf{M}, \mathbf{N}$ respectively.

Again as in previous section 4.4, we consider optimization problem 4.27. Since $g_1(\mathbf{E})$ is fixed, we just need to optimize over $g_2(\mathbf{E})$. This optimization can be reduced to local optimizations 4.29. Since $\mathbf{L}^0 = \mathbb{1}_{\mathcal{R}}^{n \times n}$ and the condition (1.3),

$$\mathbf{E}_{\mathcal{R}^c} \text{ is (entrywise) nonnegative}$$
$$\mathbf{E}_{\mathcal{R}} \text{ is (entrywise) nonpositive}$$

We can make use of Lemma 4.8 and assume $\mathbf{m}_l^{\mathcal{C}_i}$ is nonpositive/nonnegative when $i = l/i \neq l$ for all $i$. Hence using Lemma 4.30 we lower bound $g(\mathbf{v}_l\mathbf{m}_l^T)$ as described in the following section.

### 5.3.1   Lower bounding $g(\mathbf{E})$

**Lemma 5.5.** *Assume,* $1 \leq l \leq K$, $\tilde{\mathbf{D}}_{\mathcal{A}} > 0$. *Then, with probability* $1 - n\exp(-2\tilde{\mathbf{D}}_{\mathcal{A}}^2(n_l - 1))$, *we have* $g(\mathbf{v}_l\mathbf{m}_l^T) \geq \lambda(1 - q - \tilde{\mathbf{D}}_{\mathcal{A}})sum(\mathbf{v}_l\mathbf{m}_l^T)$ *for all* $\mathbf{m}_l$. *Also, if* $\mathbf{m}_l \neq 0$ *then inequality is strict.*

*Proof.* Recall that $\mathbf{m}_l$ satisfies $\mathbf{m}_l^{\mathcal{C}_i}$ is nonpositive/nonnegative when $i = l/i \neq l$ for all $i$. Define $\mathbf{X}^i := \mathbb{1}^{n_l}\mathbf{m}_l^{\mathcal{C}_i}{}^T$. We can write

$$g(\mathbf{v}_l\mathbf{m}_l^T) = \frac{1}{n_l}sum(\mathbf{X}^l) + \sum_{i=1}^{K} \lambda sum(\mathbf{X}_{\beta_{l,i}^c}^i)$$

Now assume $i \neq l$. Using Lemma 4.9 and the fact that $\beta_{l,i}$ is a randomly generated subset (with parameter $q$), with probability $1 - n_i\exp(-2\epsilon'^2 n_l)$, for all $\mathbf{X}^i$, we have,

$$sum(\mathbf{X}_{\beta_{l,i}^c}^i) \geq (r(1-q) - \epsilon')sum(\mathbf{X}^i) \tag{5.11}$$

where inequality is strict if $X^i \neq 0$. Similarly, when $i = l$ with probability at least $1 - n_l\exp(-2\epsilon'^2(n_l - 1))$, we have,

$$\frac{1}{\lambda n_l}sum(\mathbf{X}^l) + sum(\mathbf{X}_{\beta_{l,l}^c}^l) \geq \left(\frac{1}{\lambda n_l} + r(1-p_l) + \epsilon'\right)sum(\mathbf{X}^l)$$

Together,

$$g(\mathbf{v}_l\mathbf{m}_l^T) \geq \lambda \sum_{i \neq l}(r(1-q) - \epsilon')sum(\mathbf{X}^i) + \left(\frac{1}{\lambda n_l} + r(1-p_l) + \epsilon'\right)sum(\mathbf{X}^l)$$

$$\geq \lambda(r(1-q) - \epsilon')\sum_{i=1}^{K} sum(\mathbf{X}^i) = \lambda(r(1-q) - \epsilon')sum(\mathbf{v}_l\mathbf{m}_l^T) \tag{5.12}$$

Choosing $\epsilon' = \frac{\tilde{\mathbf{D}}_{\mathcal{A}}}{2}$ and using the facts that $r(1-q) - 2\tilde{\mathbf{D}}_{\mathcal{A}} \geq 0$, $r(p_l - q) - \frac{1}{\lambda n_l} - 2\tilde{\mathbf{D}}_{\mathcal{A}} \geq 0$ and using a union bound, with probability $1 - n\exp(-2\tilde{\mathbf{D}}_{\mathcal{A}}^2(n_l - 1))$, we have $g(\mathbf{v}_l\mathbf{m}_l^T) \geq 0$ and the inequality is strict when $\mathbf{m}_l \neq 0$ as at least one of the $\mathbf{X}^i$'s will be nonzero. ∎

The following lemma immediately follows from Lemma 5.5 and summarizes the main result of the section.

**Lemma 5.6.** *Let* $\tilde{\mathbf{D}}_{\mathcal{A}}$ *be as defined in* (4.1) *and assume* $\tilde{\mathbf{D}}_{\mathcal{A}} > 0$. *Then with probability* $1 - 2nK\exp(-2\tilde{\mathbf{D}}_{\mathcal{A}}^2(n_{min} - 1))$ *we have* $g(\mathbf{E}^L) > 0$ *for all nonzero feasible* $\mathbf{E}^L \in \mathcal{M}_{U}^{\perp}$.

## 5.4  The Final Step

**Lemma 5.7.** *Let $p_{min} > q$ and $\mathcal{G}$ be a random graph generated according to Model 2.1 and 2.2 with cluster sizes $\{n_i\}_{i=1}^{K}$. If $\lambda \leq (1-\epsilon)\tilde{\mathbf{\Lambda}}_{succ}$ and $\mathbf{D}_{\min} = \min\limits_{1 \leq i \leq n} r\,(p_i - q)\,n_i \geq (1+\epsilon)\frac{1}{\lambda}$, then $\left(\mathbf{L}^0, \mathbf{S}^0\right)$ is the unique optimal solution to Program 1.1 with probability $1 - \exp(-\Omega(n)) - 6n^2 \exp(-\Omega(n_{min}))$.*

*Proof.* Based on Lemma 5.4 and Lemma 5.6,
with probability $1 - cn^2 \exp(-C(r(p_{min} - q))^2 n_{min})$,

- There exists $\mathbf{W} \in \mathcal{M}_{\mathbf{U}}$ with $\|\mathbf{W}\| < 1$ such that for all feasible $\mathbf{E}^L$, $f(\mathbf{E}^L, \mathbf{W}) \geq 0$.

- For all nonzero $\mathbf{E}^L \in \mathcal{M}_{\mathbf{U}}^{\perp}$ we have $g(\mathbf{E}^L) > 0$.

Consequently based on Lemma 4.3, $(\mathbf{L}^0, \mathbf{S}^0)$ is the unique optimal of Problem 1.4.  ∎