[Reviews · NeurIPS 2014]

Submitted by Assigned_Reviewer_24

In this paper the authors analyze theoretically two common graph clustering algorithms using low rank + sparsity, showing bounds on the parameter of these methods for them to work, and they present experimental validations of the results.

The paper is very well written in general, although there are some minor typos. For instance, I think that the "about" in line 314 should be an "above". Also, it seems more reasonable to me to put subsection 3.1.1 as 3.2, and several figures are a couple of pages before the first mention of them. Besides these minor details, I really liked how the paper is written.

The problem of graph clustering is of great relevance, and in general parameters are not easy to tune, so this theoretical analysis is very important.

One thing that I found intriguing is the large gap between the optimistic and pessimistic bounds in Figure 1. I would like to see an analysis on this fact. In particular, how does this behave asymptotically?

The labeling application, although it's more a comparison between existing methods than a verification of the presented theorems, is interesting.
Summary: The paper is well written, with interesting theory, supported by experimental results, which can be further improved by analizing the gap between the two bounds.

Submitted by Assigned_Reviewer_36

The paper aims at clustering graphs with missing information. It is modeled as a convex optimization problem, along the same line as the works of Candes-et.al. about matrix completion [23] and robust PCA [24]. Additional linear constraints are considered to fit the clustering problem with missing information. The two core problems 1.1 and 1.4 in the paper are not new in clustering literature and were already studied in a series of papers as discussed in the manuscript. In this context, the proposed work attempts to extend existing results. It seems to me that the proposed contribution is okay but not enough good for NIPS. The newly theoretical results do not seem to produce new essential insights on problems 1.1. and 1.4. Experiments are also limited as they are done on small datasets.
Summary: The paper extends existing results for the clustering problem formulated as a matrix completion/robust PCA problem. It seems an interesting but limited extension of existing results, with no new significant insights and no important experimental results.

Submitted by Assigned_Reviewer_39

The paper analyzes two convex programs for graph clustering with missing data under the stochastic block model. Success conditions are provided for both programs, and failure conditions are also given for the first one. Understanding the exact performance boundary of these programs is an interesting and relevant problem. However, as mentioned by the authors, there are several existing work on the analysis of similar convex programs. It is not entirely clear how the results in this paper differentiate from existing ones, and the paper lacks a detailed comparison. For example:

(1) The work of [16] also provides success conditions for the second program. In section 3.1 the authors claim that [16] has considerably weaker bounds, but it is not clear how this is the case. An explicit comparison is needed.

(2) The work of [21] also gives success and failure conditions for the first program. Although it only considers the full observation case, the techniques seem similar.

(3) The success and failure conditions for the first program (in THM 1) do not complement each other, even if one ignores the difference between Lambda_succ and Lambda_fail. For example, what happens when D_min < 1/Lambda_succ? Does the program fail for all choices of lambda?

(4) The results seem even weaker than existing bounds in certain regimes. Suppose n_min = n/2 and r=1/2. The success condition for the second program requires, among other things, p > 1/sqrt(n); see eq.(2.4). However, if we treat missing entries as 0 and apply standard results for spectral clustering (e.g. McSherry), then we only need p > O(log(n)/n), which is order-wise better.

In terms of writing, the statements of the main theorems are quite technical and complicated, and it is difficult to interpret them. It would be better if more discussion is given.
Summary: The paper provides success and failure conditions for two convex programs for graph clustering. While technically sound, the results seem similar to existing work, and in certain regimes appear to be weaker than previous known bounds. A more detailed comparison with existing work is desired.

Submitted by Assigned_Reviewer_44

The paper proposes and analyzed a new formulation for graph clustering, extending previous works [11-20] that use of nuclear norm minimization for this problem (with minor differences). The version of stochastic block model assumed in this paper is that: nodes within the same cluster have larger probability than if they are not for all clusters, and only a random subset of the edges are observed. The improvement of this paper over earlier results is to allow exact recovery of the graph cluster structure with cluster edge density < 0.5. This is achieved by adding a few additional meaningful constraints in (1.4). The formulation seems to combine ideas in [15,16,19] so as to handle missing data and sparse "errors" at the same time.

In general, I feel that the paper contains important improvement over previous works.It has solid theoretical results (I wasn't able to verify everything in the supplementary materials, but they are believable and easy to follow) and illustrative numerical simulation. Also the paper is very clearly written. Lastly, the Mechanical Turk experiments are really nice, given that this line of work typically have weak practical impact.

Cons:
1. Providing explicit constants is really good, but a comparison with previous work in terms of the rate of recovery (ignoring constant) is also necessary. Please add a table if possible.
2. (For these graph clustering work in general) practical usefulness. Graph clusters in practice never obey these distributional assumptions. Results under weaker and more realistic conditions will be more useful. We can't blame the authors for this though, the setup (SBM) is kind of standard in graph clustering.

Minor comments:
eqn (2.1): a typo? q should be the probability that the non-cluster entries have edges.
Summary: The paper improves the existing algorithms for "nuclear norm" based graph clustering algorithms. The results are nice and a recommend acceptance of this.
Author Feedback
Author rebuttal: We thank the reviewers for their detailed comments and observations. In thinking about these, and putting together the rebuttal, we have realized that we can improve the paper by strengthening the statement of Thm 1 and providing more insight for Program 1.4

Reviewer 1:
Q: The gap between success and failure
The gap between Lambda_succ and Lambda_fail can be decreased by modifying the dual certificate W_0 in (4.20). In fact, if we change the second term of W_0 so that it is “all ones” over the intersection between the out-of-cluster and observed regions (and not only the out-of-cluster regions) then Lambda_succ^-1 in the beginning of Section 2.2 becomes 4 sqrt(n r q (1-q)). Comparing with the expression for Lambda_fail^-1, we see that the gap between Lambda_succ and Lambda_fail is at most a factor of 4.

Reviewer 2:
Q1: Experiments are on a small dataset
To the best of our knowledge, the current literature on convex-optimization-based clustering algorithms [13-17], [19], [21], does not contain “any” experiments with real data. Our experiment is on a relatively small dataset partially because how best to scale such methods to larger data sets is ongoing work. Nonetheless, the demonstration that convex-optimization-based techniques outperform k-means in certain applications is interesting and valuable.
Q2: Regarding new insights
Our analysis provides an understanding of the bounds on the regularization parameter lambda and provides a principled approach to tuning it.
Further, thanks to the observation of Reviewer 3 (regarding the very sparse regime—see below), we obtain the following practical insight: “For denser graphs (where we can run Program 1.1) it is useful to consider only the observed edges of the graph and keep the unobserved ones arbitrary. However, for very sparse graphs (where we must use Program 1.4) it is *always* better to set the unobserved entries to 0.”

Reviewer 3:
Q1: Regarding [16] having weaker bounds
For Program (1.1), scaling the edge probabilities to rp and rq would require rp > 1/2. This is weaker than working only with the observed entries which requires p > 1/2 (under appropriate conditions on Dmin). For Program (1.4), our analysis is much tighter than that of [16]. When r=1, our minimum cluster size grows as sqrt(qn), rather than sqrt(n).
Q2:Similarity of proof technique to [21]
We agree with the reviewer that our proof technique follows the same line of [21], i.e., it carefully constructs a dual certificate and shows that the objective function strictly increases under perturbations. This is a common proof technique for many convex optimization problems. Further, while [21] analyzes the fully observed case for (1.1), our work analyzes the partially observed cases for both (1.1) and (1.4). We agree that we could expand on the comparison and differentiation of our work from existing literature. Currently, our discussions in sections 1.2 and 1.3 are limited due to space constraints; but we would be happy to add expanded literature comparisons in the supplementary material.
Q3: What happens in Thm 1 when D_min < 1/lambda?
The program fails with high probability when D_min < 1/lambda. Thus, the second statement in Thm 1 can be strengthened to “If 0 < lambda < (1-epsilon)Lambda_success, then Program 1.1 succeeds… *if and only if* D_min > (1+epsilon)/lambda”
This can be seen as follows. Note that S (in (1.1)) is 0 over the unobserved entries (otherwise the objective can be strictly decreased by setting those entries to 0). Thus the subgradient of S over the unobserved entries can also be set to 0. With this observation, the RHS of (4.9) becomes lambda n_i^2 r (1-2p_i). Recalling that D_min = n_i r (2p_i-1), it follows that if Dmin < 1/lambda, the KKT condition (4.3) is violated with high probability.
Q4: Comparison with McSherry’s result in very sparse regime (p<<1)
For Program 1.1, setting the unobserved entries to 0 will not work for r < 1/2 or 1/2 < p < 1/(2r). So we gain by running the suggested program.
Turning to Program 1.4, when r=1, our results match those of McSherry. Indeed McSherry’s condition reads as n > c sqrt(np logn)/(p-q), whereas ours reads as n > c’ sqrt(np)/(p-q).
However, when r<1 Program 1.4 does behave worse than McSherry’s result (and is even orderwise worse in the setting the reviewer mentions---we thank the reviewer for this observation). The reason has to do with allowing the unobserved edges of the graph to be arbitrary.
Critical to Program 1.4 is the constraint (1.6): L_ij = S_ij when A_ij=0 (which is the only constraint involving A). With missing data: A_ij(observed)=0 with probability r(1-p) inside the clusters and r(1-q) outside the clusters. Defining pbar = 1-r(1-p) = rp+1-r and qbar = 1-r(1-p) = rq+1-r, the number of constraints in (1.6) becomes statistically equivalent to those of a “fully observed” graph where p and q are replaced by pbar and qbar. Plugging pbar and qbar into the expression n > c’ sqrt(npbar)/(pbar-qbar) for r=1 yields the weak result the reviewer refers to.
However, setting the unobserved entries to 0, yields A_ij = 0 with probability 1-rp inside the clusters and 1-rq outside them. The expression now becomes n > c’ sqrt(np/r)/(p-q), which is considerably improved. Intuitively, clustering a fully observed graph with pbar = rp+1-r and qbar = rq+1-r is much more difficult than one with rp and rq, since the links are “more noisy”.
In summary, for cluster sizes O(n), if we set the unobserved entries to 0, and replace p and q with rp and rq, then both our and McSherry's formula will allow p > O(log n/ n). This indicates that in Program 1.4 “it is always beneficial to set the unobserved entries to 0.” We would be happy to include a detailed discussion of this critical observation.

Q5: Thms are technical and complicated to parse
With translated parameters pbar = rp+1-r and qbar = rq+1-r, we will be able to simplify the expressions so that they can be more easily parsed.